# GENERALIZED NATURAL GRADIENT FLOWS IN HIDDEN CONVEX-CONCAVE GAMES AND GANS

**Andjela Mladenovic** [*]
Univ. of Montréal & Mila

**Iosif Sakos**
SUTD

**Gauthier Gidel**[†]
Univ. of Montréal & Mila

**Georgios Piliouras**
SUTD

## ABSTRACT

Game-theoretic formulations in machine learning have recently risen in prominence, whereby entire modeling paradigms are best captured as zero-sum games. Despite their popularity, however, their dynamics are still poorly understood. This lack of theory is often substantiated with painful empirical observations of volatile training dynamics and even divergence. Such results highlight the need to develop an appropriate theory with convergence guarantees that are powerful enough to inform practice. This paper studies the generalized Gradient Descent-Ascent (GDA) flow in a large class of non-convex non-concave Zero-Sum games dubbed Hidden Convex-Concave games, a class of games that includes GANs. We focus on two specific geometries: a novel geometry induced by the hidden convex-concave structure that we call the hidden mapping geometry and the Fisher information geometry. For the hidden mapping geometry, we prove global convergence under mild assumptions. In the case of Fisher information geometry, we provide a complete picture of the dynamics in an interesting special setting of team competition via invariant function analysis.

## 1 INTRODUCTION

Min-max optimization has found extensive applications in modern Machine Learning (ML) and Deep Learning. Popular application settings include Generative Adversarial Networks (GANs) (Goodfellow et al., 2014), adversarial training (Madry et al., 2018), and multi-agent reinforcement learning (Silver et al., 2017). In all of these cases, a pair of networks is typically trained towards finding an approximate equilibrium of a highly non-convex non-concave problem. However, such settings go beyond classical and well-known results in game theory for which equilibria only exist in more restrictive settings, i.e., convex-concave games (Sion et al., 1958). Unfortunately, there is no parallel guarantee if the payoff is not convex-concave, and all these applications are indeed based upon non-convex non-concave games. Even worse, many negative results occur when dealing with such payoffs: global or local minimax may not exist (Jin et al., 2020), and even if they exist, there is, in general, no "reasonable algorithm"[1] that can globally converge to any meaningful notion of local optimum (Letcher, 2021; Hsieh et al., 2020). To overcome these negative results, we analyze a specific class of non-convex non-concave games called Hidden Convex-Concave (HCC) games (Flokas et al., 2020; Gidel et al., 2021; Flokas et al., 2021) that include many Machine Learning applications such as GANs (Goodfellow et al., 2014), Adversarial Example Games (AEG) (Bose et al., 2020), or Minimax Estimation of Conditional Moment (Dikkala et al., 2020).

**Our contributions.** In the setting of HCC games, our analysis tackles unique challenges not occurring in the standard convex-concave games setting. We propose a new type of dynamics, dubbed *Natural Hidden Gradient dynamics (NHG)*, and we prove its convergence to stationary points of the HCC game. Critically, *our convergence results are global*, i.e., we do not make any assumptions about initial conditions, e.g., that the initial points belong to some local neighborhood. This novel algorithm is inspired by the generalization of gradient flows in different Banach spaces than the standard Euclidean one. Arguably, the most well-known non-Euclidean gradient is the natural gradient

---

[*]Emails of contact:{`andjela.mladenovic`,`gidelgau`}@mila.quebec,
`iosif_sakos@mymail.sutd.edu.sg`, `georgios@sutd.edu.sg`

[†]Canada CIFAR AI Chair

[1]see (Letcher, 2021, Definition 5) for a definition of "reasonable algorithm"

flow induced by the Fisher information matrix, which enjoys deep connections to the Replicator Dynamics (RD). Thus, a natural second question emerges: Can we have similar convergence guarantees for RD? In that regard, we provide a complete picture of the resulting dynamics in a setting of team competition via invariant function analysis. Specifically, we reduce the flow of the dynamics to a competition between two generalized gradients, one for each team. We show that the behavior of the dynamics in this setting is constrained by a maximal number of invariant functions, which, combined with the actual geometry of the available strategies, shows that the system can either converge or cycle. These two results showcase, both, the importance of adapting the "algorithmic geometry" to the application and the data at hand, as well as the unexplored effects of feasibility constraints on the complexity of well-known game dynamics.

## 2 Related Work

Despite the popularity of deep learning applications involving non-convex non-concave games, our understanding of their optimization dynamics and the nature of their solution concepts are still preliminary. However, this space has already witnessed a few important early works that focus on identifying new solution concepts. These solution concepts—which are also broadly applicable in general min-max games—include (local/differential) Nash equilibria (Adolphs et al., 2019; Mazumdar & Ratliff, 2019), (local/differential) Stackelberg equilibria (Fiez et al., 2020; Wang et al., 2020), local minmax (Daskalakis & Panageas, 2018), local robust points (Zhang et al., 2020), and approximate minimax theorems (Jin et al., 2020; Gidel et al., 2021). Numerous solutions concepts such as cycles (Vlatakis-Gkaragkounis et al., 2019), chaotic behavior (Cheung & Piliouras, 2019; Cheung & Tao, 2021), and computational issues (Daskalakis et al., 2021) indicate that solving min-max games, in general, might involve challenging and complex behavior.

Many algorithms have been proposed to solve restricted classes of non-convex non-concave games such as Polyak-Łojasiewicz games (Nouiehed et al., 2019; Yang et al., 2020), nonconvex-concave games (Lin et al., 2020; Ostrovskii et al., 2021; Yang et al., 2020; Kong & Monteiro, 2021), as well as classes of games inspired by variational inequalities (Mertikopoulos et al., 2019; Diakonikolas et al., 2021; Lee & Kim, 2021). However, even though they are significant advances in the understanding of non-convex non-concave game dynamics, such classes of games may not encompass games where the players are parameterized neural networks such as GANs.

While convergence in GANs has been a topic of exploration (Kodali et al., 2017; Heusel et al., 2017; Mescheder et al., 2018; Gemp & Mahadevan, 2018; Li et al., 2018; Hsieh et al., 2019; Cao & Guo, 2020), its hidden convex-concave structure has not been exploited before, and there were no theoretical global convergence guarantees in a general setting up to this date. In particular, Hsieh et al. (2019) use a lifting trick and proceed in solving a relaxation of the problem in the distribution space, while in our work we work entirely in the parameter space. With this respect, we consider Flokas et al. (2021), and Gemp & Mahadevan (2018) as the closest related works. On the one hand, in Gemp & Mahadevan (2018), the authors propose *crossing-the-curl*, a second-order technique, and provide global convergence guarantees for the Wasserstein Linear Quadratic GAN (W-LQGAN). While the W-LQGANs is a class of non-convex non-concave GANs it remains far from the GAN formulation used in practice where the discriminator and the generator are neural networks. On the other hand, the idea of Hidden Convex-Concave (HCC) games was first proposed by Flokas et al. (2021) and Gidel et al. (2021). While Flokas et al. (2021) study the Gradient Descent-Ascent (GDA) dynamics in the HCC setting, their work relies on hard-to-verify safety conditions. Our global convergence results without hard-to-test assumptions on initialization are first of their kind, to the best of knowledge.

## 3 Preliminaries

Many game-theoretic applications in machine learning often involve a specific structure where the models' payoff is a convex-concave function (e.g., minimizing Jensen-Shannon-Divergence when training GANs). To model this specific structure, we propose the following definition of *Hidden Convex-Concave (HCC) games* where intuitively, the game's payoff is a convex-concave function whose actions are parametrized by non-convex mappings.

**Definition 1** (Hidden Convex-Concave game). *A Hidden Convex-Concave game comprises a collection of payoff $(L_{\boldsymbol{x},\boldsymbol{x}'})_{\boldsymbol{x},\boldsymbol{x}'\in\mathbb{R}^d}$, a distribution $p$, and two parametrized mappings, $F : \mathbb{R}^M \times \mathbb{R}^d \to \mathbb{R}$, and $G : \mathbb{R}^N \times \mathbb{R}^{d'} \to \mathbb{R}$, such that the minimax game of interest is*

$$\min_{\boldsymbol{\theta}\in\mathbb{R}^M} \max_{\boldsymbol{\phi}\in\mathbb{R}^N} \Psi(\boldsymbol{\theta},\boldsymbol{\phi}) \quad where \quad \Psi(\boldsymbol{\theta},\boldsymbol{\phi}) = L(F_{\boldsymbol{\theta}}, G_{\boldsymbol{\phi}}) \coloneqq \mathbb{E}_{(\boldsymbol{x},\boldsymbol{x}')\sim p}[L_{\boldsymbol{x},\boldsymbol{x}'}(F_{\boldsymbol{\theta}}(\boldsymbol{x}), G_{\boldsymbol{\phi}}(\boldsymbol{x}'))]. \quad (1)$$

In this setting, while we do not expect $\Psi(\boldsymbol{\theta},\boldsymbol{\phi})$ to be convex-concave, we assume the function $L : \mathbb{F}\times\mathbb{G} \to \mathbb{R}$ to be convex-concave where $\mathbb{F}$ and $\mathbb{G}$ are, respectively, convex subsets of $\{F : \mathbb{R}^d \to \mathbb{R}\}$ and $\{G : \mathbb{R}^p \to \mathbb{R}\}$.[2] We extended Flokas et al. (2021)'s definition to now be able to include most minimax machine learning applications such as GANs or AEG.

**Example 1** (Hidden Matching Pennies (HMP) games). *Let us consider $p, q \in [0, 1]$ the probabilities of picking HEADS for the first and the second player, respectively, in a Matching Pennies game with payoff matrix $\boldsymbol{A} \in \mathbb{R}^{2\times 2}$, where $A_{i,j} = 1$ if $i = j$; $-1$, otherwise. Then the payoff of this game is defined via*

$$L_{\boldsymbol{x},\boldsymbol{x}'}(p,q) \coloneqq (1-2p)(1-2q) \quad if (p,q) \in [0,1]^2 \qquad and \qquad 0 \ otherwise. \quad (2)$$

*Now, let us consider any mappings $F : \mathbb{R}^M \times \mathbb{R}^d \to [0,1]$ and $G : \mathbb{R}^N \times \mathbb{R}^d \to [0,1]$, such that their output does not depend on the $d$-dimensional input, i.e., $F_{\boldsymbol{\theta}}(\boldsymbol{x}) = f(\boldsymbol{\theta})$, and $G_{\boldsymbol{\phi}}(\boldsymbol{x}) = g(\boldsymbol{\phi})$, $\forall \boldsymbol{x} \in \mathbb{R}^d$. The payoff $\Psi(\boldsymbol{\theta},\boldsymbol{\phi}) = L(F_{\boldsymbol{\theta}}, G_{\boldsymbol{\phi}}) \coloneqq \mathbb{E}[L_{\boldsymbol{x},\boldsymbol{x}'}(F_{\boldsymbol{\theta}}(\boldsymbol{x}), G_{\boldsymbol{\phi}}(\boldsymbol{x}'))] = (1 - 2f(\boldsymbol{\theta}))(1 - 2g(\boldsymbol{\phi}))$ defines a HCC game.*

In this example, the two agents play the typical bilinear game of Matching Pennies. However, they do not act on it directly (i.e., choose randomized actions to apply, e.g., the probability of playing HEADS). Instead, they choose the input parameters $\boldsymbol{\theta}$, and $\boldsymbol{\phi}$, which are fed into functions $f$, and $g$, respectively, whose outputs define the probability of playing HEADS for each agent.

**Example 2** (GANs). *A Generative Adversarial Network (GAN) is a minimax game where the first player, i.e., the generator, aims at learning a distribution $p_{\boldsymbol{\theta}}$ similar to a reference data distribution $p_{\text{data}}$. In practice, the reference data distribution is taken to be the empirical data distribution. Conversely, the second player, usually called the discriminator or critic, $D_{\boldsymbol{\phi}}$, tries to distinguish the distributions of $p_{\boldsymbol{\theta}}$ and $p_{\text{data}}$. The payoff, $\Psi$, of this game is defined as:*

$$\Psi(\boldsymbol{\theta},\boldsymbol{\phi}) = \mathbb{E}_{\mathbf{x}\sim p_{data}}[\log D_{\boldsymbol{\phi}}(\boldsymbol{x})] + \mathbb{E}_{\boldsymbol{x}\sim p_{\boldsymbol{\theta}}}[\log(1 - D_{\boldsymbol{\phi}}(\boldsymbol{x}))]. \quad (3)$$

*Assuming that $p_{\text{data}}$ and $p_{\boldsymbol{\theta}}$ have a density with respect to Lebesgue measure,[3] and that the support of $p_{\boldsymbol{\theta}}$ is included in the support of $p_{\text{data}}$, we can consider the distribution $p$ such that $p(\boldsymbol{x},\boldsymbol{x}') = p_{\text{data}}(\boldsymbol{x})$ if $\boldsymbol{x} = \boldsymbol{x}'$, and $0$ otherwise, and set $L_{\boldsymbol{x},\boldsymbol{x}'}(p', D) \coloneqq \log D + \frac{p'}{p_{\text{data}}(\boldsymbol{x}')}\log(1 - D)$. Thus $L_{\boldsymbol{x},\boldsymbol{x}'}$ is convex-concave for any $\boldsymbol{x}, \boldsymbol{x}' \in \mathbb{R}^d$. We have that $\Psi(\boldsymbol{\theta},\boldsymbol{\phi}) = \mathbb{E}_{(\boldsymbol{x},\boldsymbol{x}')\sim p}[L_{\boldsymbol{x},\boldsymbol{x}'}(p_{\boldsymbol{\theta}}(\boldsymbol{x}), D_{\boldsymbol{\phi}}(\boldsymbol{x}'))]$, which is a HCC game.*

A GAN formulates the generative modeling task as finding a Nash equilibrium of a minimax game. The generator of a GAN is defined as a function that aims to produce realistic data samples by transforming samples drawn from a fixed noise distribution, e.g., $\mathcal{N}(0, \boldsymbol{I}_d)$. Here, we notice that the GAN payoff is convex (actually linear) as a function of the density of the generated distribution. An alternative formulation of a GAN is a *Wasserstein GAN (WGAN)* (Arjovsky et al., 2017). It turns out that this GAN is also a HCC game.

**Example 3** (WGANs). *A Wasserstein GAN is a constrained minimax game, where the second player $D_{\boldsymbol{\phi}}$ is a $1$-Lipchitz function and where the payoff is*

$$\Psi(\boldsymbol{\theta},\boldsymbol{\phi}) = \mathbb{E}_{\mathbf{x}\sim p_{\text{data}}}[D_{\boldsymbol{\phi}}(\mathbf{x})] - \mathbb{E}_{\mathbf{x}\sim p_{\boldsymbol{\theta}}}[D_{\boldsymbol{\phi}}(\mathbf{x})]. \quad (4)$$

*Similarly, as in Example 2, the WGAN payoff can be shown to be a HCC game.*

As the last class of examples of HCC games, we present the *Adversarial Example Games (AEG)* (Bose et al., 2020).

---

[2]One can always assume $\mathbb{F}$ and $\mathbb{G}$ to be convex sets by considering their convex hulls.

[3]We make this assumption for simplicity. We can consider Radon–Nikodym derivatives for the general case.

**Example 4** (AEG). *An Adversarial Example Game is a minimax game between a generator $G_{\boldsymbol{\theta}}$ and a classifier $f$. Given samples $(\boldsymbol{x}, \mathrm{y}) \sim p_{\text{data}}$, the generator $G_{\boldsymbol{\theta}}$ aims at finding adversarial examples $\boldsymbol{x}'$ such that $\|\boldsymbol{x} - \boldsymbol{x}'\|_{\infty} \leq \epsilon$ and that $\boldsymbol{x}'$ is not classified as $\mathrm{y}$ by $f$, thus generating a distribution $p_{\boldsymbol{\theta}}$. Overall, the payoff of this game is*

$$\Psi(\boldsymbol{\theta}, \boldsymbol{\phi}) = -\mathbb{E}_{(\mathbf{x}', \mathrm{y}) \sim p_{\boldsymbol{\theta}}}[\ell(f_{\boldsymbol{\phi}}(\mathbf{x}'), \mathrm{y})] \ \ for \ \ (\mathbf{x}', \mathrm{y}) \sim p_{\boldsymbol{\theta}} \iff \boldsymbol{x}' = G_{\boldsymbol{\theta}}(\boldsymbol{x}), \ (\mathbf{x}, \mathrm{y}) \sim p_{\text{data}}, \quad (5)$$

*and where $\ell$ is the cross-entropy loss and $G_{\theta}$ is such that $\|\boldsymbol{x} - G_{\boldsymbol{\theta}}(\boldsymbol{x})\| \leq \epsilon, \forall \boldsymbol{x}$. By using a similar construction as in Example 2, we can show that* (5) *is a HCC game with respect to $p_{\boldsymbol{\theta}}$ and $f_{\boldsymbol{\phi}}$.*

In this work, we make the assumption that the minimax problem induced by $L$ admits a solution:

**Assumption 1.** *The HCC game defined by* (1) *admits a Nash equilibrium, $(F^*, G^*)$, i.e.,*

$$\min_{F \in \mathbb{F}} \max_{G \in \mathbb{G}} L(F, G) = \max_{G \in \mathbb{G}} \min_{F \in \mathbb{F}} L(F, G) = L(F^*, G^*) . \quad (6)$$

Such an assumption is relatively mild since it holds when the set $\mathbb{F}$ is compact (by definition, it is convex) (Sion et al., 1958). Note that this solution may not be achievable, i.e., we do not assume that there exists $\boldsymbol{\theta}^*$ and $\boldsymbol{\phi}^*$ such that $(F_{\boldsymbol{\theta}^*}, G_{\boldsymbol{\phi}^*}) = (F^*, G^*)$. Such sufficient conditions for the existence of a Nash equilibrium of $L : \mathbb{F} \times \mathbb{G} \to \mathbb{R}$ are discussed in detail in Gidel et al. (2021, Prop. 1), e.g., Assumption 1 holds if the parameters $\boldsymbol{\theta}$ and $\boldsymbol{\phi}$ are bounded. From a high-level perspective, this assumption is analogous to the existence of a global solution non-convex optimization. We use $(F^*, G^*)$ as a target to build a Lyapunov function of the natural hidden gradient flow.

## 3.1 NATURAL GRADIENT FLOW

In this section, we present the notion of a *natural gradient flow* (Amari, 1985; 1998). Let us consider a function $f : \mathbb{S} \to \mathbb{R}$ and a class of symmetric positive definite matrices $(\boldsymbol{P}_{\boldsymbol{\theta}})_{\boldsymbol{\theta} \in \mathbb{S}} \succ 0$ that we will refer to as *metric tensors*. The *natural gradient flow* is the flow given by the steepest descent direction (Ollivier et al., 2017) with respect to the geometry induced by the matrices $\boldsymbol{P}_{\boldsymbol{\theta}}$,

$$\dot{\boldsymbol{\theta}} = -\boldsymbol{P}_{\boldsymbol{\theta}}^{-1} \nabla f(\boldsymbol{\theta}) . \quad (7)$$

When $\boldsymbol{P}_{\boldsymbol{\theta}} = \boldsymbol{I}$, we consider the canonical Euclidean geometry and recover the standard gradient flow. One celebrated example of a natural gradient in machine learning is the natural gradient induced by the Fisher information matrix (Amari, 1998; Martens, 2020).

**Example 5** (The natural gradient flow of the Fisher information matrix). *Let us consider $P(\mathbb{X})$ the space of probability distributions on a set $\mathbb{X} \subseteq \mathbb{R}$ with the metric induced by the Kullback–Leibler (KL) divergence. Then the natural gradient flow of the Fisher information matrix is given by*

$$\dot{\boldsymbol{\theta}} = -\boldsymbol{F}_{\boldsymbol{\theta}}^{-1} \nabla f(\boldsymbol{\theta}) \quad where \quad \boldsymbol{F}_{\boldsymbol{\theta}} \coloneqq -\mathbb{E}_{\mathbf{x} \sim p_{\boldsymbol{\theta}}}[\nabla_{\boldsymbol{\theta}}^2 \log p_{\boldsymbol{\theta}}(\mathbf{x})], \ p_{\boldsymbol{\theta}} \in P(\mathbb{X}) . \quad (8)$$

*Moreover, if $n \coloneqq |\mathbb{X}| = \dim \mathbb{S}$ is finite, and if $p_{\boldsymbol{\theta}} = \boldsymbol{\theta}$, the flow* (8) *is the natural gradient flow of the Shahshahani metric (Shahshahani, 1979) induced by the metric tensors*

$$\boldsymbol{S}_{\boldsymbol{\theta}} \coloneqq \text{diag}(\tfrac{1}{\theta_1}, \dots, \tfrac{1}{\theta_n}) . \quad (9)$$

Recently, alternative natural gradient formulations have been developed using the Wasserstein distance (Li & Montúfar, 2018; Arbel et al., 2020).

## 3.2 CONNECTIONS BETWEEN REPLICATOR DYNAMICS AND GRADIENT FLOWS

The *Replicator Dynamics (RD)* are standard dynamics used in Evolutionary Game Theory and learning in games. It is, arguably, the most widely used model of evolutionary selection with multiple applications in economics, biology, and other fields. Interestingly, RD enjoys a close connection to gradient flows (see Sigmund (1984); Hofbauer et al. (1998); Harper (2009); Mertikopoulos & Sandholm (2018)). Specifically, in the case of a symmetric and linear fitness landscape, the gradient induced by the Shahshahani metric of the mean fitness is a special case of RD. Such a landscape can be formally represented by a *Potential game*. A Potential game is a $n$-player game $\mathbf{G} = (n, \mathbb{S} \coloneqq [m_1] \times \dots \times [m_n], u : \mathbb{S} \to \mathbb{R}^n)$, with payoff function $u$, characterized by the existence of a *potential function* $\Phi : \mathbb{S} \to \mathbb{R}$ that satisfies

$$\Phi(\boldsymbol{s}_i, \boldsymbol{s}_{-i}) - \Phi(\boldsymbol{s}'_i, \boldsymbol{s}_{-i}) = u_i(s_i, s_{-i}) - u_i(s'_i, s_{-i}) \quad \forall i \in [n] . \quad (10)$$

The connection between potential games and the gradient flows induced by the Shahshahani metric can be formalized with a generalization of Hofbauer et al. (1998)'s lemma.

**Proposition 1.** *The Replicator Dynamics of a potential game* **G** *with potential function* $\Phi$ *is an (extended) Shahshahani gradient in* $\text{int}(\Delta_{m_1} \times \ldots \times \Delta_{m_n})$ *having potential* $\Psi(\boldsymbol{\theta}) := \mathbb{E}_{\substack{\mathbf{s}_i \sim \boldsymbol{\theta}_i \\ i \in [n]}}[\Phi(\mathbf{s})]$.

It follows that, for potential games, RD is a gradient flow with respect to a very specific geometry. In the next section, we consider a geometry induced by the HCC structure, and which we leverage to obtain a new natural gradient flow that we call *Natural Hidden Gradient flow* (NGH).

### 3.3 The Natural Hidden Gradient flow for HCC Games

In HCC games, we assume $\Psi(\boldsymbol{\theta}, \boldsymbol{\phi}) = L(F_{\boldsymbol{\theta}}, G_{\boldsymbol{\phi}})$ (see Definition 1), and, therefore, since the entities that characterize a solution to the minimax problem are the mappings $F_{\boldsymbol{\theta}}$ and $G_{\boldsymbol{\phi}}$ instead of their *parametrizations*, $\boldsymbol{\theta}$ and $\boldsymbol{\phi}$, respectively, a natural geometry for consideration is the one defined by the $L^2$ distance in the space of $\mathbb{F} \times \mathbb{G}$. Formally, let $\boldsymbol{\theta}$ and $\boldsymbol{\theta}'$ be two parameterizations for $F$ (similarly, for $G$). The $L^2$ distance between the mappings are:

$$\|F_{\boldsymbol{\theta}'} - F_{\boldsymbol{\theta}}\|^2 := \mathbb{E}_{\boldsymbol{x} \sim p_{\boldsymbol{x}}}(F_{\boldsymbol{\theta}'}(\boldsymbol{x}) - F_{\boldsymbol{\theta}}(\boldsymbol{x}))^2 \quad \text{and} \quad \|G_{\boldsymbol{\phi}} - G_{\boldsymbol{\phi}'}\|^2 := \mathbb{E}_{\boldsymbol{x} \sim p_{\boldsymbol{x}'}}(G_{\boldsymbol{\phi}'}(\boldsymbol{x}') - G_{\boldsymbol{\phi}}(\boldsymbol{x}'))^2.$$

where $p_{\boldsymbol{x}}$ and $p_{\boldsymbol{x}'}$ are the marginal of $p_{\boldsymbol{x}, \boldsymbol{x}'}$. We can then derive the metric tensors of this geometry.

**Proposition 2** (Metric tensors of the model space). *Under mild regularity assumptions, we have that, for any* $\boldsymbol{\theta} \in \mathbb{R}^M$,

$$\|F_{\boldsymbol{\theta}+\delta\boldsymbol{\theta}} - F_{\boldsymbol{\theta}}\|^2 = \langle \delta\boldsymbol{\theta}, \boldsymbol{A}_{\boldsymbol{\theta}} \delta\boldsymbol{\theta} \rangle + o(\|\delta\boldsymbol{\theta}\|^2) \quad \text{where} \quad \boldsymbol{A}_{\boldsymbol{\theta}} := \mathbb{E}_{\boldsymbol{x} \sim p_{\boldsymbol{x}}}[\nabla_{\boldsymbol{\theta}} F_{\boldsymbol{\theta}}(\boldsymbol{x}) \nabla_{\boldsymbol{\theta}} F_{\boldsymbol{\theta}}(\boldsymbol{x})^{\mathsf{T}}]. \quad (11)$$

Consequently, $\boldsymbol{A}_{\boldsymbol{\theta}}$ defines a metric on the parameter space in which the "distance" between two values, $\boldsymbol{\theta}$, and $\boldsymbol{\theta}'$, corresponds to the $L^2$ distance between $F_{\boldsymbol{\theta}}$ and $F_{\boldsymbol{\theta}'}$. We can, thus, construct the corresponding natural gradient flow.

**Proposition 3** (Natural Hidden Gradient dynamics). *The flow induced by the geometry* (11) *is*

$$\begin{aligned} \dot{\boldsymbol{\theta}} &= -\boldsymbol{A}_{\boldsymbol{\theta}}^{\dagger} \mathbb{E}_{(\boldsymbol{x}, \boldsymbol{x}') \sim p}[\nabla_{\boldsymbol{\theta}} L_{\boldsymbol{x}, \boldsymbol{x}'}(F_{\boldsymbol{\theta}}(\boldsymbol{x}), G_{\boldsymbol{\phi}}(\boldsymbol{x}'))] \\ \dot{\boldsymbol{\phi}} &= \boldsymbol{B}_{\boldsymbol{\phi}}^{\dagger} \mathbb{E}_{(\boldsymbol{x}, \boldsymbol{x}') \sim p}[\nabla_{\boldsymbol{\phi}} L_{\boldsymbol{x}, \boldsymbol{x}'}(F_{\boldsymbol{\theta}}(\boldsymbol{x}), G_{\boldsymbol{\phi}}(\boldsymbol{x}'))] \end{aligned} \quad \text{(D1)}$$

*where* $\boldsymbol{A}_{\boldsymbol{\theta}} := \mathbb{E}_{\boldsymbol{x} \sim p_{\boldsymbol{x}}}[\nabla_{\boldsymbol{\theta}} F_{\boldsymbol{\theta}}(\boldsymbol{x}) \nabla_{\boldsymbol{\theta}} F_{\boldsymbol{\theta}}(\boldsymbol{x})^{\mathsf{T}}]$ *and* $\boldsymbol{B}_{\boldsymbol{\phi}} := \mathbb{E}_{\boldsymbol{x}' \sim p_{\boldsymbol{x}'}}[\nabla_{\boldsymbol{\phi}} G_{\boldsymbol{\phi}}(\boldsymbol{x}) \nabla_{\boldsymbol{\phi}} G_{\boldsymbol{\phi}}(\boldsymbol{x})^{\mathsf{T}}]$ *and* $\boldsymbol{C}^{\dagger}$ *denotes the* pseudo-inverse *of a matrix* $\boldsymbol{C}$.

## 4 Convergence of the Natural Hidden Gradient flow

In this section, we propose a new type of dynamics, dubbed *Natural Hidden Gradient dynamics (NHG)*, and in the following two theorems, we prove their convergence in HCC games and GANs. At the heart of our analysis lies the construction of a proper Lyapunov function that measures the distance from the game's equilibrium point. By proving that the Lyapunov function is proper, i.e., monotonic, we will prove our proposed dynamics are approaching a Nash Equilibrium.

### 4.1 Warm-up: A single datapoint

In this section, as a warm-up, we will consider the single datapoint case. In this case, (1) is,

$$\Psi(\boldsymbol{\theta}, \boldsymbol{\phi}) = L(F_{\boldsymbol{\theta}}(\boldsymbol{x}), G_{\boldsymbol{\phi}}(\boldsymbol{x}')) \quad (12)$$

where $L : \mathbb{R} \times \mathbb{R} \to \mathbb{R}$ is convex-concave (see Example 1). In this case, since the mapping $F_{\boldsymbol{\theta}}$ and $G_{\boldsymbol{\phi}}$ are evaluated at a single point, one can simplify the notation and consider $F_{\boldsymbol{\theta}}(\boldsymbol{x}) = f(\boldsymbol{\theta})$ and $G_{\boldsymbol{\phi}}(\boldsymbol{x}') = g(\boldsymbol{\phi})$ where $f : \mathbb{R}^M \to \mathbb{R}$ and $g : \mathbb{R}^N \to \mathbb{R}$ are real-valued mappings that do not depend on an input $\boldsymbol{x}$ or $\boldsymbol{x}'$. This situation is already non-trivial since, as illustrated in Example 1, it can correspond to a non-convex non-concave parametrization of the Matching Pennies game. In order to solve this game we consider the Natural gradient of the metric defined in Proposition 2.

**Proposition 4.** *In the uni-dimensional case, the Natural Hidden Gradient flow D1 takes the form of*

$$\dot{\boldsymbol{\theta}} = -\frac{\nabla_{\boldsymbol{\theta}} L(f(\boldsymbol{\theta}), g(\boldsymbol{\phi}))}{\|\nabla_{\boldsymbol{\theta}} f(\boldsymbol{\theta})\|^2} \quad \text{and} \quad \dot{\boldsymbol{\phi}} = \frac{\nabla_{\boldsymbol{\phi}} L(f(\boldsymbol{\theta}), g(\boldsymbol{\phi}))}{\|\nabla_{\boldsymbol{\phi}} g(\boldsymbol{\phi})\|^2}. \quad \text{(D2)}$$

Using Proposition 4, it is relatively straightforward to show that the distance to the optimum is a Lyapunov function for the Natural Hidden Gradient flow.

**Theorem 1.** *Let $\Psi$ be the payoff of an HCC game* (12) *and consider the dynamics* (D2)*. Then,*

$$V(\boldsymbol{\theta}, \boldsymbol{\phi}) := \tfrac{1}{2}(f(\boldsymbol{\theta}) - f^*)^2 + \tfrac{1}{2}(g(\boldsymbol{\phi}) - g^*)^2 \tag{13}$$

*is a Lyapunov function, i.e., it is positive, non-increasing, and null if and only if it evaluated at a game solution. Moreover, if $L$ is strictly convex-concave, we have that $V$ is decreasing and that any limit point $(\boldsymbol{\theta}, \boldsymbol{\phi})$ satisfies $\nabla_{\boldsymbol{\theta}} L(f(\boldsymbol{\theta}), g(\boldsymbol{\phi})) = \nabla_{\boldsymbol{\phi}} L(f(\boldsymbol{\theta}), g(\boldsymbol{\phi})) = 0$.*

In order to see this, one can compute the time derivative of $V$. After some elementary computations, it follows $\dot{V}(\boldsymbol{\theta}, \boldsymbol{\phi}) = -(f(\boldsymbol{\theta}) - f^*)\frac{\partial L(f, g(\boldsymbol{\phi}))}{\partial f}\Big|_{f=f(\boldsymbol{\theta})} + (g(\boldsymbol{\phi}) - g^*)\frac{\partial L(f(\boldsymbol{\theta}), g)}{\partial g}\Big|_{g=g(\boldsymbol{\phi})}$, and, thus, by the convex-concavity of $L$, we have that $\dot{V} \leq 0$. However, generalizing this theorem to HCC games with non-constant mappings (with respect to $\boldsymbol{x}$) is non-trivial since we drastically used the simplicity of the mappings (Proposition 4) to simplify the expression of the flow. In the next section, we propose to extend our convergence analysis to finite sum HCC games.

### 4.2 THE GENERAL FINITE SUM CASE

In this section, we consider a finite-sum version of the HCC games as they appear in Definition 1. In this case, the payoff is defined as

$$\Psi(\boldsymbol{\theta}, \boldsymbol{\phi}) := \frac{1}{nm} \sum_{(i,j)\in[m]\times[b]} L_{i,j}(F_{\boldsymbol{\theta}}(\boldsymbol{x}_i), G_{\boldsymbol{\phi}}(\boldsymbol{x}'_j)) =: L\left((F_{\boldsymbol{\theta}}(\boldsymbol{x}_i))_{i\in[m]}, (G_{\boldsymbol{\phi}}(\boldsymbol{x}'_j))_{j\in[n]}\right), \tag{14}$$

where the function $L : \mathbb{R}^m \times \mathbb{R}^n \to \mathbb{R}$ is assumed to be convex-concave. We note $L_{i,j} := L_{\boldsymbol{x}_i, \boldsymbol{x}'_j}$ for compactness. Let us recall that we assumed the existence of a Nash equilibrium $(F^*, G^*)$ for the minimax problem (Assumption 1). In this situation, the Natural gradient defined in Proposition 3 has the following form:

$$\dot{\boldsymbol{\theta}} = -\frac{A_{\boldsymbol{\theta}}^{\dagger}}{nm} \sum_{(i,j)\in[m]\times[n]} \nabla_{\boldsymbol{\theta}} L_{i,j}(F_{\boldsymbol{\theta}}(\boldsymbol{x}_i), G_{\boldsymbol{\phi}}(\boldsymbol{x}'_j)), \quad \dot{\boldsymbol{\phi}} = \frac{B_{\boldsymbol{\phi}}^{\dagger}}{nm} \sum_{(i,j)\in[m]\times[n]} \nabla_{\boldsymbol{\phi}} L_{i,j}(F_{\boldsymbol{\theta}}(\boldsymbol{x}_i), G_{\boldsymbol{\phi}}(\boldsymbol{x}'_j)), \tag{D3}$$

where $\boldsymbol{A}_{\boldsymbol{\theta}} := \frac{1}{m} \sum_{i=1}^{m} \nabla_{\boldsymbol{\theta}} F_{\boldsymbol{\theta}}(\boldsymbol{x}_i) \nabla_{\boldsymbol{\theta}} F_{\boldsymbol{\theta}}(\boldsymbol{x}_i)^{\mathsf{T}}$ and $\boldsymbol{B}_{\boldsymbol{\phi}} := \frac{1}{n} \sum_{j=1}^{n} \nabla_{\boldsymbol{\phi}} G_{\boldsymbol{\phi}}(\boldsymbol{x}'_j) \nabla_{\boldsymbol{\phi}} G_{\boldsymbol{\phi}}(\boldsymbol{x}'_j)^{\mathsf{T}}$.

We will generalize the Lyapunov function considered in the uni-dimensional case (13). The idea is to consider the $L^2$ distance between $(F_{\boldsymbol{\theta}}(\boldsymbol{x}_i), G_{\boldsymbol{\phi}}(\boldsymbol{x}'_j))_{i,j}$ and $(F^*(\boldsymbol{x}_i), G^*(\boldsymbol{x}'_j))_{i,j}$ as our Lyapunov function $V$. However, in order to prove our result, we will need the following technical assumption.

**Assumption 2.** *For any $\boldsymbol{\theta} \in \mathbb{R}^M$ and $\boldsymbol{\phi} \in \mathbb{R}^N$, we have that the families of vectors $(\nabla_{\boldsymbol{\theta}} F_{\boldsymbol{\theta}}(\boldsymbol{x}_i))_{i\in[m]}$ and $(\nabla_{\boldsymbol{\phi}} G_{\boldsymbol{\phi}}(\boldsymbol{x}'_j))_{j\in[n]}$ are linearly independent.*

When the models are overparametrized, e.g., $M > m$ and $N > n$, this assumption is relatively mild since it can be insured by a small perturbation of the considered vectors. In practice, it suggests regularizing the matrices $\boldsymbol{A}_{\boldsymbol{\theta}}$ and $\boldsymbol{B}_{\boldsymbol{\phi}}$ by adding $\epsilon \cdot \boldsymbol{I}_d$ which is a standard way to stabilize methods requiring matrix inversions.

**Theorem 2.** *Let $\Psi$ be the payoff of a finite-sum HCC game given by* (14) *and consider the game dynamics in* (D3)*. Under Assumption 1, Assumption 2, we have that the quantity*

$$V(\boldsymbol{\theta}, \boldsymbol{\phi}) := \frac{1}{2n} \sum_{i=1}^{n} (F_{\boldsymbol{\theta}}(\boldsymbol{x}_i) - F^*(\boldsymbol{x}_i))^2 + \frac{1}{2m} \sum_{j=1}^{m} (G_{\boldsymbol{\phi}}(\boldsymbol{x}'_j) - G^*(\boldsymbol{x}'_j))^2 \tag{15}$$

*is a Lyapunov function, i.e., is positive, non-increasing and null if and only if evaluated at a game solution. Moreover, if $L$ is strictly convex-concave, $V$ is decreasing as long as $(\boldsymbol{\theta}, \boldsymbol{\phi}) \neq (\boldsymbol{\theta}^*, \boldsymbol{\phi}^*)$ and if $L$ is a $\mu$-strongly convex-concave function we have that $V$ is decreasing exponentially as $V(\boldsymbol{\theta}, \boldsymbol{\phi}) = V(\boldsymbol{\theta}_0, \boldsymbol{\phi}_0) \exp(-\mu t)$.*

We showed that, in the overparametrized regime, if we assume not to encounter any singular matrices $A_{\boldsymbol{\theta}}$ and $B_{\boldsymbol{\phi}}$ along the trajectory, then, preconditioning low dimensional gradient of $\boldsymbol{\theta}$ and $\boldsymbol{\phi}$ can behave like doing gradient update on $F$ and $G$ to leverage the hidden-convex-cave structure of

the the payoff. We do not know how to recover the gradients updates on $F$ and $G$ in the non-overparametrized regime. It is a great open question that we consider outside of the scope of this paper as we focus on understanding convergence in minimax games for deep learning models (that are over-parametrized). The proofs of Theorem 1 and Theorem 2 are in §A.3.

Regarding the practicability of the method described in (D3), efficient approximations of preconditioning, such as the K-FAC algorithm, were proposed (Martens, 2020; Li & Montúfar, 2018) and used to train large models on Imagenet and CIFAR (Martens et al., 2021; Arbel et al., 2020).

## 5 CHARACTERIZATIONS OF REPLICATOR DYNAMICS IN HCC GAMES

In this section, we consider a specific instance of HMP games (cf. Example 1), dubbed *2-Team HMP games* or the *XOR-XOR games*. Although the possibility of cycling orbits for RD in such games was established before (Piliouras & Schulman, 2018), in this section, we show stronger results. Specifically, as we prove in subsection 5.2, by enforcing restrictions to the game's parameters, it is possible to affect the game's outcome, e.g., we may deviate from the well-known cyclic behavior in the unrestricted setting, and moreover, enforce divergence away from the game's original equilibrium and convergence to novel fixed points. We completely characterize the geometry of possible limit cycles in such a restricted setting by exploiting intuitions developed via the connection between RD and the Shahshahani information geometry. Specifically, we show the dynamics are controlled by invariant functions, which correspond to weighted sums of the cross-entropy of the current mixed strategies of opposing members relative to the uniform, equilibrium strategies (19). For the complete proofs of this section, we refer the interested reader to §A.4.

### 5.1 2-TEAM HIDDEN MATCHING PENNIES GAMES

We introduce the *2-Team HMP game*, $\mathbf{G} = (n := n_1 + n_2, \mathbb{S} := \{0,1\}^n, u : \mathbb{S} \to \mathbb{R}^n)$, between $n_1 + n_2$ members divided into two teams. The first team, *team 1*, consists of $n_1$ members, while the second team, *team 2*, consists of $n_2$ members. The payoff function $u$ for a strategy profile $\boldsymbol{s} := (\boldsymbol{s}_1, \boldsymbol{s}_2) \in \mathbb{S}$ is given by

$$u_{k,i}(\boldsymbol{s}) := \frac{(-1)^{k-1}}{n_k}(1 - 2 \cdot \mathbf{1}_{\mathrm{XOR}(\mathrm{s}_1) = \mathrm{XOR}(\mathrm{s}_2)}), \ k \in [2], \ i \in [n_k]. \tag{16}$$

where $\mathrm{XOR}(\boldsymbol{s}_k) = 1$ if $|\{s_{k,i} \mid s_{k,i} = 1\}|$ is odd, and 0, otherwise. Given a mixed-strategy profile, $(\boldsymbol{\theta}, \mathbf{1} - \boldsymbol{\theta})$, $\boldsymbol{\theta} := (\boldsymbol{\theta}_k)_{k \in [2]} \in [0,1]^n$, the expected payoff of the $i$-th member of team $k$ is given by

$$\Psi_{k,i}(\boldsymbol{\theta}) := \mathbb{E}_{\substack{s_{k,i} \sim \mathrm{Ber}(\theta_{k,i}) \\ k \in [2], \ i \in [n_k]}}[u_{k,i}(\mathbf{s})] = \frac{(-1)^{k-1}}{n_k}(1 - 2f(\boldsymbol{\theta}_1))(1 - 2g(\boldsymbol{\theta}_2)) \tag{17}$$

where $f(\boldsymbol{\theta}_1) := \mathbb{E}_{\mathbf{s}_1 \sim \mathrm{Ber}(\boldsymbol{\theta}_1)}[\mathrm{XOR}(\mathbf{s}_1)]$ and $g(\boldsymbol{\theta}_2) := \mathbb{E}_{\mathbf{s}_2 \sim \mathrm{Ber}(\boldsymbol{\theta}_2)}[\mathrm{XOR}(\mathbf{s}_2)]$. Notice that, since each member of a given team aims at maximizing the same payoff, $\mathbf{G}$ is a Hidden Matching Pennies game (Example 1) with hidden mappings $f(\boldsymbol{\theta}_1)$, and $g(\boldsymbol{\theta}_2)$,

$$\min_{\boldsymbol{\theta}_2} \max_{\boldsymbol{\theta}_1} \Psi(\boldsymbol{\theta}_1, \boldsymbol{\theta}_2) := \sum_{i=1}^{n_1} \Psi_{1,i}(\boldsymbol{\theta}) = (1 - 2f(\boldsymbol{\theta}_1))(1 - 2g(\boldsymbol{\theta}_2)). \tag{18}$$

It is not difficult to prove that the RD exhibit cyclic behavior in this setting. The following theorem provides a fine-grained characterization of the dynamics of the HCC game (18) where we show that the trajectories lie on the intersection of level sets of invariant functions.

**Theorem 3.** *Consider the Replicator Dynamics of* $\mathbf{G}$. *Given any interior initial condition, the resulting orbit is a cycle that satisfies the following* $n_1 + n_2 - 1$ *independent invariant functions:*

$$V_{i_1, i_2}(\boldsymbol{\theta}) = \sum_{k=1}^{2} n_k[\log(\theta_{k,i_k}) + \log(1 - \theta_{k,i_k})], \ i_k \in [n_k], k \in [2]. \tag{19}$$

### 5.2 REPLICATOR DYNAMICS OF HIDDEN MATCHING PENNIES IN A RESTRICTED SETTING

Next, we introduce restrictions in the range of each member's strategies in $\mathbf{G}$ such that $\theta_{k,i} \in \mathbb{S}_{k,i} := [\alpha_{k,i}, \beta_{k,i}] \subseteq [0,1]$, $\forall k \in [2]$, $i \in [n_k]$. These restrictions reflect in RD as halts, i.e., $\dot{\boldsymbol{\theta}}_{k,i} = 0$, every

time the strategy of the $i$-th member of the $k$-th team exceeds those bounds. To ease our notation, we let $\mathbb{S}_k := \{i \in [n_k] \mid \frac{1}{2} \in \mathbb{S}_{k,i}\}$, $k \in [2]$ denote all the members of the $k$-th team for which $(\frac{1}{2}, \frac{1}{2})$ is an allowed mixed strategy, and, to simplify this part of the analysis, we also make the following mild assumption on the initialization of the dynamics:

**Assumption 3.** *For all $k \in [2]$ and $i \in [n_k]$, $\theta_{k,i}(0) \in \mathbb{S}_{k,i} \subset (0, 1)$.*

A significant observation is that for any mixed-strategy profile $(\boldsymbol{\theta}, \mathbf{1} - \boldsymbol{\theta})$, $\boldsymbol{\theta} := (\boldsymbol{\theta}_1, \boldsymbol{\theta}_2) \in [0, 1]^n$, if there exists some $i_k \in [n_k]$, $\forall k \in [2]$ such that $\theta_{i_k} = \frac{1}{2}$, then $\boldsymbol{\theta}$ is an equilibrium point; in fact, it is not difficult to see that these are the only interior equilibrium points of **G**, which implies that an interior equilibrium point is reachable if and only if $\mathbb{S}_k \neq \emptyset$, $\forall k \in [2]$. In our first result, we prove that if the Nash Equilibrium of the unrestricted case is not reachable by both teams due to the constraints, then the dynamics converges to a point which only depends on those restrictions.

**Theorem 4.** *Under Assumption 3, if $\mathbb{S}_k = \emptyset$, $\forall k \in [2]$, the restricted RD of **G** converge to*

$$
\underbrace{\begin{aligned}
\theta_{1,i}^* &= \begin{cases} \alpha_{1,i}, & \text{if } \beta_{1,i} < \frac{1}{2} \\ \beta_{1,i}, & \text{otherwise} \end{cases} \quad i \in [n_1] \\
\theta_{2,i}^* &= \begin{cases} \beta_{2,i}, & \text{if } \beta_{2,i} < \frac{1}{2} \\ \alpha_{2,i}, & \text{otherwise} \end{cases} \quad i \in [n_2]
\end{aligned}}_{\text{if } |\mathbb{S}| \text{ is even}}
\quad \text{or} \quad
\underbrace{\begin{aligned}
\theta_{1,i}^* &= \begin{cases} \beta_{1,i}, & \text{if } \beta_{1,i} < \frac{1}{2} \\ \alpha_{1,i}, & \text{otherwise} \end{cases} \quad i \in [n_1] \\
\theta_{2,i}^* &= \begin{cases} \alpha_{2,i}, & \text{if } \beta_{2,i} < \frac{1}{2} \\ \beta_{2,i}, & \text{otherwise} \end{cases} \quad i \in [n_2]
\end{aligned}}_{\text{if } |\mathbb{S}| \text{ is odd}}
\tag{20}
$$

*where $\mathbb{S} := \{(k, i) \mid k \in [2], i \in [n_k], \alpha_{k,i} > \frac{1}{2}\}$.*

The key idea behind this result is that at any given time $t \geq 0$, the direction of a strategy $\theta_{k,i}(t)$ only depends on whether $|\mathbb{S}|$, the total number of members who can access the uniform strategy, is odd or even. Thus, we can decouple the evolution of the dynamics of each of the members and analyze its behavior separately. If an equilibrium point of the unrestricted setting is reachable by both teams, i.e., $\mathbb{S}_k \neq \emptyset$, $\forall k \in [2]$, then the dynamics converge to an invariant set whose degrees of freedom depend on the number of members who have access to the uniform strategy, $(\frac{1}{2}, \frac{1}{2})$.

**Theorem 5.** *Under Assumption 3, if $\mathbb{S}_k \neq \emptyset$, $\forall k \in [2]$, then the restricted RD of **G** converge to an invariant set defined by the $|\mathbb{S}_1| + |\mathbb{S}_2| - 1$ independent invariant functions, $V_{i_1,i_2}(\boldsymbol{\theta})$, $i_k \in \mathbb{S}_k$, $\forall k \in [2]$, where $V_{i_1,i_2}(\boldsymbol{\theta})$ is given as in (19).*

One can prove this result by partitioning the time based on the set of members that have halted. The analysis of $V_{i_1,i_2}(\boldsymbol{\theta})$ in each time-partition is almost trivial, and the result follows by a continuity argument on $V_{i_1,i_2}(\boldsymbol{\theta})$. These results depict how the behavior of RD depends on the parameter space of the game. Notably, we show that if no equilibrium point is feasible, the RD converge to a point described entirely by the strategy space restrictions. On the other hand, if an equilibrium point is feasible, we prove the existence of a maximal number of invariant functions, with close connections to the KL divergence. The latter does not merely show that the RD cycle in this setting, but that they actually converge to an invariant set with *specific degrees of freedom*, and which we characterize. For parameterizations that visualize the behavior of RD in such restricted settings, see §B.1.

## 6 EXPERIMENTAL RESULTS

**Toy multi-dimensional case.** As a first experiment, we consider the HCC objective

$$
\Psi(\boldsymbol{\theta}, \boldsymbol{\phi}) = L(F_{\boldsymbol{\theta}}, G_{\boldsymbol{\phi}}) := F_{\boldsymbol{\theta}}^{\mathsf{T}} M G_{\boldsymbol{\phi}} + \tfrac{\lambda}{2}(\|F_{\boldsymbol{\theta}} - \tfrac{1}{3}\|^2 - \|G_{\boldsymbol{\phi}} - \tfrac{1}{3}\|^2),
$$

where $M$ is the payoff matrix of the Rock-Paper-Scissors game (see e.g. Gidel et al. (2021)). We note $F_{\boldsymbol{\theta}}(x_i) = [F_{\boldsymbol{\theta}}]_i$, $i \in [3]$, and $\frac{1}{3}$, i.e., the uniform distribution, is the game's equilibrium. The mappings $F$ and $G$ are 2-layer MLP with 130 parameters and GELU non-linearities (Hendrycks & Gimpel, 2016). In Figure 1, we depict a comparison between the performance of GDA and NHG dynamics on the task of solving $\min_{\boldsymbol{\theta}} \max_{\boldsymbol{\phi}} \Psi(\boldsymbol{\theta}, \boldsymbol{\phi})$. We remark, the NHG dynamics converges smoothly (Figure 1 *(Right)*), compared to GDA (Figure 1 *(Center)*), which fail to converge. The value of the Lyapunov function of the game described is depicted in (Figure 1 *(Left)*).

**GANs.** For our second experiment, we implement the NHG dynamics to train a GAN. We consider a synthetic experiment to learn a sine wave sampled uniformly from 0 to $\pi$ with 1024 observations.

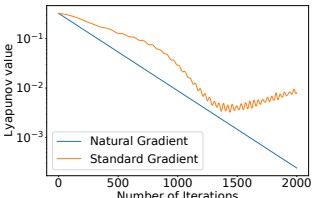 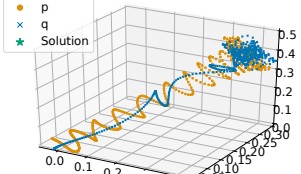 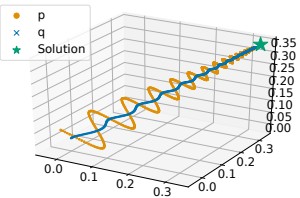

Figure 1: A comparison of GDA *(Center)* and NHG dynamics *(Right)* in the task of solving $\min_{\theta} \max_{\phi} \Psi(\theta, \phi)$ where $p = f(\theta)$, and $q = g(\phi)$. In NHG, the Lyapunov function *(Left)* is monotonically decreasing, as opposed to GDA.

Regarding the GAN, we consider a Flow-GAN architecture where our generator, $G$, is a Real NVP consisting of 8 coupling layers (Dinh et al., 2017; 2014), and the discriminator is a 4-layer MLP with 256-128-64-1 output features. We adapt K-FAC (Martens & Grosse, 2015) as an approximator to compute NHG, with a learning rate of $10^{-4}$, selected using the standard parameter optimizer package Optuna (Akiba et al., 2019). We used gradient clipping with gradient clip to a maximum norm of 1 on, both the discriminator and the generator, to stabilize the optimization.

As observed in (Figure 2 *(Center)*), by iteration 200, the GAN optimized using K-FAC learns the true sinusoid almost perfectly and converges to a better Wasserstein-1 score than the conventional GDA (Figure 2 *(Right)*). The experiments reveal that our approach provides good performance and convergence guarantees. However, we observe instabilities during training, and lack of convergence in certain instances, which stay in line with the empirical observations regarding the difficulty of GAN training, and the instability of matrix inversions close to singularities. We remark that this experiment goes slightly beyond the theoretical results: while in theory, we assume a natural gradient flow on a "full-batch", our experiments are based on discrete stochastic updates, and an approximation of the pseudo-inverses of the matrices (see (D1)) using K-FAC. Thus, this experiment acts as a proof-of-concept rather than a large-scale comparison between NHG and GDA. The details of both experiments can be found within the source-code files included with this work.

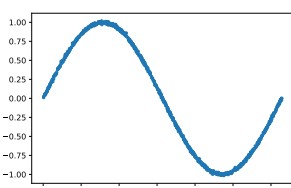 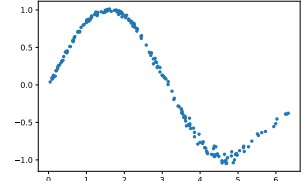 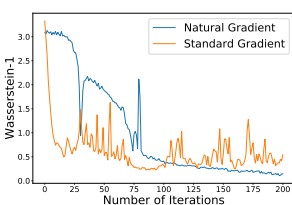

Figure 2: A plot of the real sinusoid sampled on the interval from 0 to $2\pi$ *(Left)*. GAN generated samples for NHG dynamics *(Center)*. The performance of GDA vs. NHG dynamics on GANs as measured by Wasserstein-1 distance *(Right)*.

# 7 DISCUSSION

We proposed a novel version of Gradient Descent Ascent dynamics in Hidden Convex-Concave games, a subset of non-convex non-concave games. In this class of games, which includes GANs, the utility is convex-concave in the function space. Still, training happens in the parameter space, where the mappings between the input and output are non-convex non-concave functions. We explored the dynamics of gradient flows induced by different Banach spaces (e.g., the Fischer information geometry) that led to the discovery of Lyapunov functions suited to these geometries. Our analysis of the convergence of our proposed type of dynamics, Natural Hidden Gradient (NHG) dynamics, uses ideas from Game Theory and Dynamical Systems. We proved global convergence guarantees for NHG in HCC games to local stationary points via Lyapunov function analysis in the finite-sum case. To the best of our knowledge, such a non-local convergence result in HCC games and GAN-like settings, is one of the first of its kind. We also show promising experimental results on practical GANs using NHG and standard gradient approximation techniques such as KFAC. We are aware that the current formulation of NHG may be challenging to scale up to large neural networks because of the tensor pseudo-inversion step that is part of the dynamics. Investigating experimentally novel versions of Gradient Descent Ascent dynamics with a richer set of experiments and developing techniques to scale our results for larger neural networks is a natural direction for our future work.

## STATEMENT OF REPRODUCIBILITY

We made sure to provide sufficient details to ensure the reproducibility of our results. The complete proofs of the theoretical results can be found in §A, and all the assumptions have been stated and are referenced in each statement. We provide details regarding the experimental results, such as code language, required libraries, and parametrization to execute and reproduce the experiments in section 6 and §B. We also include the source code files and the necessary input files in the supplementary material that accompanies this work.

## ACKNOWLEDGEMENTS

The authors would like to acknowledge Joey Bose for his help on the GANs experiments.

This research/project is supported in part by the National Research Foundation, Singapore under its AI Singapore Program (AISG Award No: AISG2-RP-2020-016), NRF 2018 Fellowship NRF-NRFF2018-07, NRF2019-NRF-ANR095 ALIAS grant, grant PIE-SGP-AI-2020-01, AME Programmatic Fund (Grant No. A20H6b0151) from the Agency for Science, Technology and Research (A*STAR) and Provost's Chair Professorship grant RGEPPV2101. This work is supported by the Canada CIFAR AI Chair Program and an IVADO grant.

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

## A  OMITTED PROOFS

### A.1  OMITTED PROOFS OF SECTION 3.1

**Proposition A.1.** *Let us consider $P(\mathbb{X})$ the space of probability distributions on a set $\mathbb{X} \subseteq \mathbb{R}$ with the metric induced by the Kullback–Leibler (KL) divergence. If $n := |\mathbb{X}| = \dim \mathbb{S}$, and $p_{\boldsymbol{\theta}} := \boldsymbol{\theta} \in P(\mathbb{X})$, then the natural gradient flow of the Fisher information matrix in (8) is the natural gradient flow of the Shahshahani metric induced by the metric tensors in (9).*

*Proof.* All we need to show is that $\boldsymbol{F_\theta} = \boldsymbol{S_\theta}$, $\forall \boldsymbol{\theta} \in \mathbb{S}$. Simply, note that $\forall i, j \in [n]$:

$$
(\boldsymbol{F_\theta})_{i,j} = -\mathbb{E}_{\boldsymbol{x} \sim p_{\boldsymbol{\theta}}} \left[ \frac{\partial^2 \log p_{\boldsymbol{\theta}}(\mathrm{x})}{\partial \theta_i \partial \theta_j} \right] = -\sum_{x \in \mathbb{X}} \frac{\partial^2 \log p_{\boldsymbol{\theta}}(\mathrm{x})}{\partial \theta_i \partial \theta_j} p_{\boldsymbol{\theta}}(x)
$$

$$
= -\sum_{x \in \mathbb{X}} \frac{\partial^2 \log \theta_x}{\partial \theta_i \partial \theta_j} \theta_x = \sum_{x \in \mathbb{X}} \frac{\delta_{x,i} \delta_{x,j}}{\theta_x} = \frac{\delta_{i,j}}{\theta_i} = (\boldsymbol{S_\theta})_{i,j} \ .
$$

$\square$

### A.2  OMITTED PROOFS OF SECTION 3.2

Let $\mathbf{G} = (n, \mathbb{S} := [m_1] \times \ldots \times [m_n], u : \mathbb{S} \to \mathbb{R}^n)$ be a $n$-player (exact) potential game with payoff function $u$ and potential $\Phi : \mathbb{S} \to \mathbb{R}$, and let $\mathbb{M} := \Delta_{m_1} \times \ldots \times \Delta_{m_n}$ be the mixed-strategy space of $\mathbf{G}$. We are going to consider the RD of $\mathbf{G}$ in $\operatorname{int} \mathbb{M} = \operatorname{int} \Delta_{m_1} \times \ldots \times \operatorname{int} \Delta_{m_n}$ as given in the following proposition.

**Proposition A.2.** *The Replicator Dynamics of $\mathbf{G}$ in $\operatorname{int} \mathbb{M}$ are given by the following dynamical system of equations:*

$$
\dot{\theta}_{i,j} := \theta_{i,j}(\mathbb{E}_{\substack{\mathrm{s}_k \sim \boldsymbol{\theta}_k \\ k \in [n]}}[u_i(s_j, \mathbf{s}_{-i})] - \mathbb{E}_{\substack{\mathrm{s}_k \sim \boldsymbol{\theta}_k \\ k \in [n]}}[u_i(\mathbf{s})]) = \theta_{i,j} \left( \frac{\partial \Psi(\boldsymbol{\theta})}{\partial \theta_{i,j}} - \Psi(\boldsymbol{\theta}) \right) \tag{D4}
$$

*where $\Psi(\boldsymbol{\theta}) := \mathbb{E}_{\substack{\mathrm{s}_i \sim \boldsymbol{\theta}_i \\ i \in [n]}}[\Phi(\mathbf{s})]$ is the the expected potential function of the game.*

*Proof.* First note that

$$
\frac{\partial \Psi(\boldsymbol{\theta})}{\partial \theta_{i,j}} = \sum_{\boldsymbol{s}} \prod_{k \neq i} \theta_{k,s_k} \Phi(\boldsymbol{s}) \frac{\partial \theta_{i,s_i}}{\partial \theta_{i,j}} = \sum_{\boldsymbol{s}} \prod_{k \neq i} \theta_{k,s_k} \Phi(\boldsymbol{s}) \delta_{j,s_i}
$$

$$
= \sum_{\boldsymbol{s}_{-i}} \prod_{k \neq i} \theta_{k,s_k} \Phi(j, \boldsymbol{s}_{-i}) = \mathbb{E}_{\substack{\mathrm{s}_i \sim \boldsymbol{\theta}_i \\ i \in [n]}}[\Phi(j, \mathbf{s}_{-i})] \ .
$$

From which follows that

$$\dot{\theta}_{i,j} = \theta_{i,j}(\mathbb{E}_{\substack{s_k \sim \boldsymbol{\theta}_k \\ k \in [n]}}[u_i(j, \mathbf{s}_{-i}) - u_i(\mathbf{s})]) = \theta_{i,j}(\mathbb{E}_{\substack{s_k \sim \boldsymbol{\theta}_k \\ k \in [n]}}[\Phi(j, \mathbf{s}_{-i}) - \Phi(\mathbf{s})])$$

$$= \theta_{i,j}\left(\frac{\partial \Psi(\boldsymbol{\theta})}{\partial \theta_{i,j}} - \Psi(\boldsymbol{\theta})\right).$$

$\square$

Next, let us consider a point $\boldsymbol{\theta} \in \text{int } \mathbb{M}$, and note that the following proposition holds:

**Proposition A.3.** *For any $\boldsymbol{\theta} \in \text{int } \mathbb{M} = \text{int } \Delta_{m_1} \times \ldots \times \text{int } \Delta_{m_n}$, the tangent plane of $\text{int } \mathbb{M}$ at $\boldsymbol{\theta}$ is*

$$T_{\boldsymbol{\theta}}(\text{int } \mathbb{M}) = \{\boldsymbol{\mu} \in \mathbb{R}^{m_1} \times \ldots \times \mathbb{R}^{m_n} \mid \sum_j \mu_{i,j} = 0\}. \tag{21}$$

*Proof.* We first note that $\text{int } \mathbb{M}$ is an open-subset of the surface $\mathbb{D} := \{\boldsymbol{\mu} \in \mathbb{R}^{m_1} \times \ldots \times \mathbb{R}^{m_n} \mid U_i(\boldsymbol{\mu}) = 1\}$, where $U_i(\boldsymbol{\mu}) = \sum_j \mu_{i,j} = 1$; hence, $T_{\boldsymbol{\theta}}(\text{int } \mathbb{M}) = T_{\boldsymbol{\theta}}\mathbb{D}$, up to isomorphism. Let $\boldsymbol{r}(t)$ be a smooth curve in $\mathbb{D}$ with $\boldsymbol{r}(0) = \boldsymbol{\theta}$. Then, by definition, $U_i(\boldsymbol{r}(t)) = 1$ for all $i$, and, by differentiating with respect to $t$, we have

$$\nabla U_i(\boldsymbol{r}(t)) \cdot \frac{d}{dt}\boldsymbol{r}(t) = \frac{d}{dt}U_i(r(t)) = \frac{d}{dt}1 = 0.$$

However, $\frac{d}{dt}\boldsymbol{r}(t)$ is tangent to $\mathbb{D}$ for all $t$; hence, $\frac{d}{dt}\boldsymbol{r}(t)\big|_{t=0} \in T_{\boldsymbol{\theta}}\mathbb{D}$. It follows that $\nabla U_i(\boldsymbol{r}(0)) = \nabla U_i(\boldsymbol{\theta})$ is normal to $T_{\boldsymbol{\theta}}\mathbb{D}$ for all $i$, and ,thus, we have

$$T_{\boldsymbol{\theta}}\mathbb{D} \subseteq \{\boldsymbol{\mu} \in \mathbb{R}^{m_1} \times \ldots \times \mathbb{R}^{m_n} \mid \langle \nabla U_i(\boldsymbol{\theta}), \boldsymbol{\mu} \rangle = 0\} = \{\boldsymbol{\mu} \in \mathbb{R}^{m_1} \times \ldots \times \mathbb{R}^{m_n} \mid \sum_j \mu_{i,j} = 0\}.$$

Finally, notice that $\dim\{\boldsymbol{\mu} \in \mathbb{R}^{m_1} \times \ldots \times \mathbb{R}^{m_n} \mid \sum_j \mu_{i,j} = 0\} = \dim \mathbb{D}$, which implies the proposition.

$\square$

**Proposition 1.** *The Replicator Dynamics of a potential game $\mathbf{G}$ with potential function $\Phi$ is an (extended) Shahshahani gradient in $\text{int}(\Delta_{m_1} \times \ldots \times \Delta_{m_n})$ having potential $\Psi(\boldsymbol{\theta}) := \mathbb{E}_{\substack{s_i \sim \boldsymbol{\theta}_i \\ i \in [n]}}[\Phi(\mathbf{s})]$.*

*Proof.* Let $\boldsymbol{\theta} \in \text{int } \mathbb{M} := \text{int } \Delta_{m_1} \times \ldots \times \text{int } \Delta_{m_n}$. Then for every $\boldsymbol{\xi} \in T_{\boldsymbol{\theta}}(\text{int } \mathbb{M})$ we have:

$$\langle \dot{\boldsymbol{\theta}}, \boldsymbol{\xi} \rangle_{\boldsymbol{\theta}} = \sum \frac{1}{\theta_{i,j}}\dot{\theta}_{i,j}\xi_{i,j} = \sum \left(\frac{\partial \Psi(\boldsymbol{\theta})}{\partial \theta_{i,j}} - \Psi(\boldsymbol{\theta})\right)\xi_{i,j} = \sum \frac{\partial \Psi(\boldsymbol{\theta})}{\partial \theta_{i,j}}\xi_{i,j} - \Psi(\boldsymbol{\theta})\sum \xi_{i,j}$$

$$= \sum \frac{\partial \Psi(\boldsymbol{\theta})}{\partial \theta_{i,j}}\xi_{i,j} = \nabla\Psi(\boldsymbol{\theta}) \cdot \boldsymbol{\xi}.$$

Hence, by definition, we have that $\nabla\Psi(\boldsymbol{\theta}) = \dot{\boldsymbol{\theta}}$, where the Del operator is defined with respect to the Shahshahani metric, which implies the lemma.

$\square$

Now, let us consider a different class of games, which we define as *2-Team Zero-Sum games*.

**Definition 2.** *A 2-Team Zero-Sum game,* $\mathbf{G} = (m := m_1 + m_2, \mathbb{S} := [n_1] \times \ldots \times [n_m], u : \mathbb{S} \to \mathbb{R}^m)$, *is a game between* $m$ *players with strategy-space* $\mathbb{S}$ *whose utility function,* $u$, *satisfies the following:*

$$u_i(\boldsymbol{s}) = \begin{cases} \frac{1}{m_1}\Phi(\boldsymbol{s}), & \text{if } i \in [m_1] \\ -\frac{1}{m_2}\Phi(\boldsymbol{s}), & \text{otherwise} \end{cases} \quad \forall \boldsymbol{s} \in \mathbb{S} \tag{22}$$

*for some function* $\Phi : \mathbb{S} \to \mathbb{R}$.

We are going to, collectively, refer to the first $m_1$ players of a 2-Team Zero-Sum game, $\mathbf{G} = (m := m_1 + m_2, \mathbb{S} := [n_1] \times \ldots \times [n_m], u : \mathbb{S} \to \mathbb{R}^m)$, as team 1 and to the rest of them as team 2. Notice that if we consider each team as a single player, then the two teams are playing a 2-Player Zero-Sum game, $\mathbf{G}' = (2, \mathbb{S}' := \mathbb{S}_1 \times \mathbb{S}_2, u' : \mathbb{S}' \to \mathbb{R}^2)$ where $\mathbb{S}_1 = [n_1] \times \ldots \times [n_{m_1}], \mathbb{S}_2 = [n_{m_1+1} \times \ldots \times n_m]$, and the utility function $u'_k$ is the total utility of the corresponding team $k$, i.e. $\forall \boldsymbol{s} \in \mathbb{S}$ we have

$$u'_1(\boldsymbol{s}) = \sum_{i=1}^{m_1} \frac{1}{m_1}\Phi(\boldsymbol{s}) = \Phi(\boldsymbol{s}) \quad \text{and} \quad u'_2(\boldsymbol{s}) = -\sum_{i=m_1+1}^{m} \frac{1}{m_2}\Phi(\boldsymbol{s}) = -\Phi(\boldsymbol{s}). \tag{23}$$

Note that, by definition, the 2-Team Hidden Matching Pennies game (see section 5) is a a 2-Team Zero-Sum game. Let us consider the Replicator Dynamics of $\mathbf{G}$ given by the following dynamical system of equations:

$$\dot{\theta}_{i,j} := \theta_{i,j}(\mathbb{E}_{\mathbf{s}_k \sim \boldsymbol{\theta}_k}[u_i(j, \mathbf{s}_{-i})] - \mathbb{E}_{\mathbf{s}_k \sim \boldsymbol{\theta}_k}[u_i(\mathbf{s})])$$

$$= \begin{cases} \dfrac{1}{m_1}\theta_{i,j}\mathbb{E}_{\mathbf{s}_k \sim \boldsymbol{\theta}_k}[\Phi(j, \mathbf{s}_{-i}) - \Phi(\mathbf{s})], & \text{if } i \in [m_1] \\ -\dfrac{1}{m_2}\theta_{i,j}\mathbb{E}_{\mathbf{s}_k \sim \boldsymbol{\theta}_k}[\Phi(j, \mathbf{s}_{-i}) - \Phi(\mathbf{s})], & \text{otherwise} \end{cases} \tag{D5}$$

where, as before, we can rewrite (D5) in terms of $\Psi(x) := \mathbb{E}_{S_k \sim x_k}[\Phi(S)]$ as

$$\dot{\theta}_{i,j} = \begin{cases} \dfrac{1}{m_1}\theta_{i,j}\left(\dfrac{\partial\Psi(\boldsymbol{\theta})}{\partial\theta_{i,j}} - \Psi(\boldsymbol{\theta})\right), & \text{if } i \in [m_1] \\ -\dfrac{1}{m_2}\theta_{i,j}\left(\dfrac{\partial\Psi(\boldsymbol{\theta})}{\partial\theta_{i,j}} - \Psi(\boldsymbol{\theta})\right), & \text{otherwise}. \end{cases} \tag{D6}$$

Notice that the Replicator equations of each team are similar to the Replicator equations of a Potential game. In fact, it's not difficult to prove the following lemma.

**Lemma 1.** *Let* $\mathbf{G} = (m := m_1 + m_2, \mathbb{S} := [n_1] \times \ldots \times [n_m], u : \mathbb{S} \to \mathbb{R}^m)$ *be a 2-Team Zero-Sum game. If we assume the strategies of team 2 to be time-invariant, then the Replicator Dynamics of* $\mathbf{G}$, *given by* (D5) *(or* (D6), *equivalently), is a* ($m_1$-scaled) Shahshahani gradient in $\text{int}\,\mathbb{M} := \text{int}\,\Delta_{n_1} \times \ldots \times \text{int}\,\Delta_{n_m}$ with potential $\Psi'_1(\boldsymbol{\theta}) = \Psi \circ \Pi_1(\boldsymbol{\theta})$ where $\Psi(\boldsymbol{\theta}) := \mathbb{E}_{\mathbf{s}_i \sim \boldsymbol{\theta}_i}[\Phi(\mathbf{s})]$, and $\Pi_1 : \text{int}\,\mathbb{M} \to \text{int}\,\mathbb{M}$ is the natural projection*

$$\Pi_{k,i,j}(\boldsymbol{\theta}) = \begin{cases} \boldsymbol{\theta}_{i,j}, & \text{if } i \text{ in team } k \\ 0, & \text{otherwise}. \end{cases} \tag{24}$$

*Likewise, if we assume the strategies of team 1 to be time-invariant, then the Replicator Dynamics of* $\mathbf{G}$ *is a* ($m_2$-scaled) Shahshahani gradient in $\text{int}\,M$ with potential $\Psi'_2 = -\Psi \circ \Pi_2(\boldsymbol{\theta})$.

*Proof.* This proof is similar to the proof of Proposition 1, but we need to perform the correct projection before applying the definition of a gradient flow. Let $\boldsymbol{\theta} \in \text{int}\,\mathbb{M}$. Without any loss of the generality, let us assume that the strategies of team 2 are time-invariant, i.e., $\dot{\theta}_{i,j} = 0$ for all $i \geq m_1$. Then, for every $\boldsymbol{\xi} \in \text{T}_{\boldsymbol{\theta}}(\text{int}\,\mathbb{M})$ we have

$$\langle \dot{\boldsymbol{\theta}}, \boldsymbol{\xi} \rangle_{\boldsymbol{\theta}} = \sum \frac{m_1}{\theta_{i,j}} \dot{\theta}_{i,j} \xi_{i,j} = \sum_{i=1}^{m_1} \sum_j \frac{m_1}{e\boldsymbol{\theta}_{i,j}} \frac{1}{m_1} \theta_{i,j} \left( \frac{\partial \Psi(\boldsymbol{\theta})}{\partial \theta_{i,j}} - \Psi(\boldsymbol{\theta}) \right) \xi_{i,j}$$

$$= \sum_{i=1}^{m_1} \sum_j \frac{\partial \Psi(\boldsymbol{\theta})}{\partial \theta_{i,j}} \xi_{i,j} - \Psi(\boldsymbol{\theta}) \sum_{i=1}^{m_1} \sum_j \xi_{ij} = \sum_{i=1}^{m_1} \sum_j \frac{\partial \Psi(\boldsymbol{\theta})}{\partial \theta_{i,j}} \xi_{i,j}$$

$$= \sum \frac{\partial \Psi(\Pi_1(\boldsymbol{\theta}))}{\partial \theta_{i,j}} \xi_{i,j} = \nabla \Psi'_1(\boldsymbol{\theta}) \cdot \boldsymbol{\xi} .$$

Hence, by definition, we have that $d\Psi'_1(\boldsymbol{\theta}) = \dot{\boldsymbol{\theta}}^\flat$, which implies the theorem.

$\square$

**Proposition 2** (Metric tensors of the model space). *Under mild regularity assumptions, we have that, for any $\boldsymbol{\theta} \in \mathbb{R}^M$,*

$$\|F_{\boldsymbol{\theta}+\delta\boldsymbol{\theta}} - F_{\boldsymbol{\theta}}\|^2 = \langle \delta\boldsymbol{\theta}, \boldsymbol{A}_{\boldsymbol{\theta}} \delta\boldsymbol{\theta} \rangle + o(\|\delta\boldsymbol{\theta}\|^2) \quad where \quad \boldsymbol{A}_{\boldsymbol{\theta}} := \mathbb{E}_{\boldsymbol{x} \sim p_{\boldsymbol{x}}}[\nabla_{\boldsymbol{\theta}} F_{\boldsymbol{\theta}}(\boldsymbol{x}) \nabla_{\boldsymbol{\theta}} F_{\boldsymbol{\theta}}(\boldsymbol{x})^\mathsf{T}] . \quad (11)$$

*Proof.* The assumption on $F$ we need to prove this results are the following:

- $\boldsymbol{\theta} \mapsto F_{\boldsymbol{\theta}}(x)$ is almost surely differentiable, i.e., almost surely (in $\boldsymbol{x}$),

$$F_{\boldsymbol{\theta}+\delta\boldsymbol{\theta}}(\boldsymbol{x}) = F_{\boldsymbol{\theta}}(\boldsymbol{x}) + \langle \nabla F_{\boldsymbol{\theta}}(\boldsymbol{x}), \delta\boldsymbol{\theta} \rangle + \|\delta\boldsymbol{\theta}\| f(\boldsymbol{x}, \boldsymbol{\theta}) , \; \forall \boldsymbol{\theta} \in \mathbb{R}^M , \quad (25)$$

  where $f(\boldsymbol{x}, \delta\boldsymbol{\theta}) \to_{\delta\boldsymbol{\theta} \to 0} 0$ almost surely in $\boldsymbol{x}$.

- For small enough $\delta\theta \in \mathbb{R}^M$, The remainder of the Taylor expansion of $F_{\boldsymbol{\theta}}$ has a finite variance, i.e.,

$$\mathbb{E}_{x \sim p_x}[f(\boldsymbol{x}, \delta\boldsymbol{\theta})^2] < +\infty . \quad (26)$$

A consequence of these two assumption is that (by dominated convergence theorem and Jensen's inequality)

$$\mathbb{E}_{x \sim p_x}[|f(\boldsymbol{x}, \delta\boldsymbol{\theta})|] \to_{\delta\boldsymbol{\theta} \to 0} 0 \quad and \quad \mathbb{E}_{x \sim p_x}[f(\boldsymbol{x}, \delta\boldsymbol{\theta})^2] \to_{\delta\boldsymbol{\theta} \to 0} 0 . \quad (27)$$

Let us now prove the desired property. We start by doing a Taylor expansion of $\boldsymbol{\theta} \mapsto F_{\boldsymbol{\theta}}(x)$,

$$F_{\boldsymbol{\theta}+\delta\boldsymbol{\theta}}(\boldsymbol{x}) = F_{\boldsymbol{\theta}}(\boldsymbol{x}) + \langle \nabla F_{\boldsymbol{\theta}}(\boldsymbol{x}), \delta\boldsymbol{\theta} \rangle + \|\delta\boldsymbol{\theta}\| f(\boldsymbol{x}, \boldsymbol{\theta}) .$$

Then we have that

$$\|F_{\boldsymbol{\theta}+\delta\boldsymbol{\theta}} - F_{\boldsymbol{\theta}}\|^2 = \mathbb{E}_{\boldsymbol{x} \sim p_{\boldsymbol{x}}}[(\langle \nabla F_{\boldsymbol{\theta}}(\boldsymbol{x}), \delta\boldsymbol{\theta} \rangle + f(\boldsymbol{x}, \boldsymbol{\theta}) \|\delta\boldsymbol{\theta}\|)^2]$$

$$= \mathbb{E}_{\boldsymbol{x} \sim p_{\boldsymbol{x}}}[(\nabla F_{\boldsymbol{\theta}}(\boldsymbol{x})^\mathsf{T} \delta\boldsymbol{\theta})^2] + o(\delta\boldsymbol{\theta})^2$$

$$= \mathbb{E}_{\boldsymbol{x} \sim p_{\boldsymbol{x}}}[\delta\boldsymbol{\theta}^\mathsf{T} \nabla F_{\boldsymbol{\theta}}(\boldsymbol{x}) \nabla F_{\boldsymbol{\theta}}(\boldsymbol{x})^\mathsf{T} \delta\boldsymbol{\theta}] + o(\delta\boldsymbol{\theta})^2$$

$$= \delta\boldsymbol{\theta}^\mathsf{T} \mathbb{E}_{\boldsymbol{x} \sim p_{\boldsymbol{x}}}[\nabla F_{\boldsymbol{\theta}}(\boldsymbol{x}) \nabla F_{\boldsymbol{\theta}}(\boldsymbol{x})^\mathsf{T}] \delta\boldsymbol{\theta} + o(\delta\boldsymbol{\theta})^2 ;$$

which concludes the proof.

$\square$

**Proposition 3** (Natural Hidden Gradient dynamics). *The flow induced by the geometry* (11) *is*

$$\dot{\boldsymbol{\theta}} = -\boldsymbol{A}_{\boldsymbol{\theta}}^\dagger \mathbb{E}_{(\boldsymbol{x}, \boldsymbol{x}') \sim p}[\nabla_{\boldsymbol{\theta}} L_{\boldsymbol{x}, \boldsymbol{x}'}(F_{\boldsymbol{\theta}}(\boldsymbol{x}), G_{\boldsymbol{\phi}}(\boldsymbol{x}'))]$$

$$\dot{\boldsymbol{\phi}} = \boldsymbol{B}_{\boldsymbol{\phi}}^\dagger \mathbb{E}_{(\boldsymbol{x}, \boldsymbol{x}') \sim p}[\nabla_{\boldsymbol{\phi}} L_{\boldsymbol{x}, \boldsymbol{x}'}(F_{\boldsymbol{\theta}}(\boldsymbol{x}), G_{\boldsymbol{\phi}}(\boldsymbol{x}'))] \quad (D7)$$

*where* $\boldsymbol{A}_{\boldsymbol{\theta}} := \mathbb{E}_{\boldsymbol{x} \sim p_{\boldsymbol{x}}}[\nabla_{\boldsymbol{\theta}} F_{\boldsymbol{\theta}}(\boldsymbol{x}) \nabla_{\boldsymbol{\theta}} F_{\boldsymbol{\theta}}(\boldsymbol{x})^\mathsf{T}]$ *and* $\boldsymbol{B}_{\boldsymbol{\phi}} := \mathbb{E}_{\boldsymbol{x}' \sim p_{\boldsymbol{x}'}}[\nabla_{\boldsymbol{\phi}} G_{\boldsymbol{\phi}}(\boldsymbol{x}) \nabla_{\boldsymbol{\phi}} G_{\boldsymbol{\phi}}(\boldsymbol{x})^\mathsf{T}]$ *and* $\boldsymbol{C}^\dagger$ *denotes the* pseudo-inverse *of a matrix* $\boldsymbol{C}$.

*Proof.* The proposition directly follows from Proposition 2 applied to the definition of (7). However, in order to convey more intuition we could use the definition that the natural gradient flow of the objective function $f$ induced by the distance $d$ if

$$\dot{\boldsymbol{\theta}} = \arg\min_{\delta\boldsymbol{\theta}} \lim_{\lambda\to 0} \frac{1}{\lambda} f(\boldsymbol{\theta} + \lambda\delta\boldsymbol{\theta}) + \frac{1}{2\lambda^2} d^2(F_{\boldsymbol{\theta}+\lambda\delta\boldsymbol{\theta}}, F_{\boldsymbol{\theta}}).$$

By using the $L^2$ distance and noting that when $\lambda \to 0$ we have $f(\boldsymbol{\theta}+\lambda\delta\boldsymbol{\theta}) = f(\boldsymbol{\theta})+\lambda\nabla f(\boldsymbol{\theta})^\intercal\delta\boldsymbol{\theta} + o(\lambda)$ and $\frac{1}{2\lambda^2}\|F_{\boldsymbol{\theta}+\lambda\delta\boldsymbol{\theta}} - F_{\boldsymbol{\theta}}\|^2 = \frac{1}{2}\langle\delta\boldsymbol{\theta}, \boldsymbol{A}_{\boldsymbol{\theta}}\delta\boldsymbol{\theta}\rangle + o(1)$ we have that the RHS of the equation above is

$$\arg\min_{\delta\boldsymbol{\theta}} \nabla f(\boldsymbol{\theta})^\intercal\delta\boldsymbol{\theta} + \frac{1}{2}\langle\delta\boldsymbol{\theta}, \boldsymbol{A}_{\boldsymbol{\theta}}\delta\boldsymbol{\theta}\rangle;$$

which is minimized for $\delta\boldsymbol{\theta} = -\boldsymbol{A}_{\boldsymbol{\theta}}^\dagger\nabla f(\boldsymbol{\theta})$.

$\square$

### A.3 OMITTED PROOFS OF SECTION 4

**Proposition 4.** *In the uni-dimensional case, the Natural Hidden Gradient flow D1 takes the form of*

$$\dot{\boldsymbol{\theta}} = -\frac{\nabla_{\boldsymbol{\theta}} L(f(\boldsymbol{\theta}),g(\boldsymbol{\phi}))}{\|\nabla_{\boldsymbol{\theta}} f(\boldsymbol{\theta})\|^2} \qquad and \qquad \dot{\boldsymbol{\phi}} = \frac{\nabla_{\boldsymbol{\phi}} L(f(\boldsymbol{\theta}),g(\boldsymbol{\phi}))}{\|\nabla_{\boldsymbol{\phi}} g(\boldsymbol{\phi})\|^2}. \tag{D8}$$

*Proof.* For the uni-dimensional case, $\boldsymbol{A}_{\boldsymbol{\theta}} = \nabla f(\boldsymbol{\theta})\nabla f(\boldsymbol{\theta})^\intercal$ is a rank-1 matrix that projects any vector in the direction of $\nabla f(\boldsymbol{\theta})$ and scales it by $\|\nabla f(\boldsymbol{\theta})\|^2$. Thus, by definition of the pseudo-inverse, we have that for any vector $\boldsymbol{u} \in \mathbb{R}^d$:

$$\boldsymbol{A}_{\boldsymbol{\theta}}^\dagger\boldsymbol{u} = \frac{\nabla f(\boldsymbol{\theta})\langle\boldsymbol{u}, \nabla f(\boldsymbol{\theta})\rangle}{\|\nabla f(\boldsymbol{\theta})\|^4}.$$

Finally, notice that

$$\nabla_{\boldsymbol{\theta}} L(f(\boldsymbol{\theta}),g(\boldsymbol{\phi})) = \nabla f(\boldsymbol{\theta})\frac{\partial(f,g(\boldsymbol{\phi}))}{\partial f}\bigg|_{f=f(\boldsymbol{\theta})},$$

which leads to the stated proposition.

$\square$

**Theorem 1.** *Let $\Psi$ be the payoff of an HCC game (12) and consider the dynamics (D2). Then,*
$$V(\boldsymbol{\theta},\boldsymbol{\phi}) := \tfrac{1}{2}(f(\boldsymbol{\theta}) - f^*)^2 + \tfrac{1}{2}(g(\boldsymbol{\phi}) - g^*)^2 \tag{13}$$
*is a Lyapunov function, i.e., it is positive, non-increasing, and null if and only if it evaluated at a game solution. Moreover, if $L$ is strictly convex-concave, we have that $V$ is decreasing and that any limit point $(\boldsymbol{\theta},\boldsymbol{\phi})$ satisfies $\nabla_{\boldsymbol{\theta}} L(f(\boldsymbol{\theta}),g(\boldsymbol{\phi})) = \nabla_{\boldsymbol{\phi}} L(f(\boldsymbol{\theta}),g(\boldsymbol{\phi})) = 0$.*

*Proof.* By taking the time derivative of $V(\boldsymbol{\theta},\boldsymbol{\phi})$, we get

$$\dot{V}(\boldsymbol{\theta},\boldsymbol{\phi}) = -(\nabla f(\boldsymbol{\theta})(f(\boldsymbol{\theta})-f^*))^\intercal \frac{\nabla f(\boldsymbol{\theta})}{\|\nabla f(\boldsymbol{\theta})\|^2}\frac{\partial L(f,g(\boldsymbol{\phi}))}{\partial f}\bigg|_{f=f(\boldsymbol{\theta})}$$
$$+ (\nabla g(\boldsymbol{\phi})(g(\boldsymbol{\phi})-g^*))^\intercal \frac{\nabla g(\boldsymbol{\phi})}{\|\nabla g(\boldsymbol{\phi})\|^2}\frac{\partial L(f(\boldsymbol{\theta}),g)}{\partial g}\bigg|_{g=g(\boldsymbol{\phi})}$$
$$= -(f(\boldsymbol{\theta})-f^*)\frac{\partial L(f,g(\boldsymbol{\phi}))}{\partial f}\bigg|_{f=f(\boldsymbol{\theta})} + (g(\boldsymbol{\phi})-g^*)\frac{\partial L(f(\boldsymbol{\theta}),g)}{\partial g}\bigg|_{g=g(\boldsymbol{\phi})} \leq 0$$

where the last inequality holds because $L$ is convex-concave. Specifically, let $L : \mathbb{R}^M \times \mathbb{R}^N \to \mathbb{R}$ be any differentiable convex-concave function with saddle point $(f^*, g^*)$, then we have, respectively

$$-\langle f - f^*, \nabla_f L(f,g) \rangle \leq L(f^*, g) - L(f, g) \quad \text{and} \quad \langle g - g^*, \nabla_g L(f,g) \rangle \leq L(f,g) - L(f, g^*).$$

By adding the two inequalities, we get

$$- \langle f - f^*, \nabla_f L(f,g) \rangle + \langle g - g^*, \nabla_g L(f,g) \rangle \leq L(f^*, g) - L(f, g^*) \leq 0 \tag{28}$$

where the last inequality holds because $(f^*, g^*)$ is a saddle point. Hence, by definition, $L$ is a Lyapunov function.

Finally, when $L$ is strictly convex-concave, if there exists a limit point of $(\boldsymbol{\theta}, \boldsymbol{\phi})$ that is not a point where $\nabla_{\boldsymbol{\theta}} L(f(\boldsymbol{\theta}), g(\boldsymbol{\phi})) = 0$ and $\nabla_{\boldsymbol{\phi}} L(f(\boldsymbol{\theta}), g(\boldsymbol{\phi})) = 0$ then by strict convex-concavity we get that $V$ should decrease by a "significant enough amount" to create a contradiction with the fact that $V$ does converge.

$\square$

Next, before proving the general case of Theorem 1, we'll have to prove the following lemma:

**Lemma 2.** *Let $(\boldsymbol{u}_i)_{i \in [n]}$ be a linearly independent family of vectors on $\mathbb{R}^d$. Then, the matrix*

$$\boldsymbol{A} := \sum_{i=1}^n \boldsymbol{u}_i \boldsymbol{u}_i^\intercal. \tag{29}$$

*is the* Gram matrix *of the family $(\boldsymbol{u}_i)_{i \in [n]}$ and and we have that $\langle \boldsymbol{u}_i, \boldsymbol{A}^\dagger \boldsymbol{u}_j \rangle = \delta_{i,j}, \; \forall i, j \in [n]$.*

*Proof.* Let us introduce the matrix $\boldsymbol{P} := [\boldsymbol{u}_1, \ldots, \boldsymbol{u}_n]^\intercal$; we can easily verify that $\boldsymbol{A} = \boldsymbol{P}^\intercal \boldsymbol{A}$. Let $\boldsymbol{P} = \boldsymbol{U} \boldsymbol{D} \boldsymbol{V}^\intercal$ be the SVD decomposition of $\boldsymbol{P}$ and, thus, $\boldsymbol{A}^\dagger = \boldsymbol{V} (\boldsymbol{D}^\intercal \boldsymbol{D})^\dagger \boldsymbol{V}^\intercal$ where $\boldsymbol{D}^\intercal \boldsymbol{D}$ is a diagonal matrix. Then, we have

$$\langle \boldsymbol{u}_i, \boldsymbol{A}^\dagger \boldsymbol{u}_k \rangle = \langle \boldsymbol{P}^\intercal \boldsymbol{e}^{(i)}, \boldsymbol{A}^\dagger \boldsymbol{P}^\intercal \boldsymbol{e}^{(j)} \rangle = \langle \boldsymbol{V} \boldsymbol{D}^\intercal \boldsymbol{U}^\intercal \boldsymbol{e}^{(i)}, \boldsymbol{V} (\boldsymbol{D}^\intercal \boldsymbol{D})^\dagger \boldsymbol{V}^\intercal \boldsymbol{V} \boldsymbol{D}^\intercal \boldsymbol{U}^\intercal \boldsymbol{e}^{(j)} \rangle$$
$$= \langle \boldsymbol{e}^{(i)}, \boldsymbol{U} \boldsymbol{D} (\boldsymbol{D}^\intercal \boldsymbol{D})^\dagger \boldsymbol{D}^\intercal \boldsymbol{U}^\intercal \boldsymbol{e}^{(j)} \rangle.$$

To conclude this lemma, we just need to notice that the matrix $\boldsymbol{D}$ is a matrix with non-zero entries on the diagonal and, since we assumed that the vectors in $(\boldsymbol{u}_i)_{i \in [n]}$ are linearly independent, we have that $D_{i,i} > 0, \; \forall i \in [n]$. Thus, by a direct computation, we get that

$$\boldsymbol{D} (\boldsymbol{D}^\intercal \boldsymbol{D})^\dagger \boldsymbol{D}^\intercal = \boldsymbol{I}_n$$

which leads to $\langle \boldsymbol{u}_i, \boldsymbol{A}^\dagger \boldsymbol{u}_i \rangle = \delta_{i,j}$.

$\square$

**Theorem 2.** *Let $\Psi$ be the payoff of a finite-sum HCC game given by* (14) *and consider the game dynamics in* (D3). *Under Assumption 1, Assumption 2, we have that the quantity*

$$V(\boldsymbol{\theta}, \boldsymbol{\phi}) := \frac{1}{2n} \sum_{i=1}^n (F_{\boldsymbol{\theta}}(\boldsymbol{x}_i) - F^*(\boldsymbol{x}_i))^2 + \frac{1}{2m} \sum_{j=1}^m (G_{\boldsymbol{\phi}}(\boldsymbol{x}_j') - G^*(\boldsymbol{x}_j'))^2 \tag{15}$$

*is a Lyapunov function, i.e., is positive, non-increasing and null if and only if evaluated at a game solution. Moreover, if $L$ is strictly convex-concave, $V$ is decreasing as long as $(\boldsymbol{\theta}, \boldsymbol{\phi}) \neq (\boldsymbol{\theta}^*, \boldsymbol{\phi}^*)$ and if $L$ is a $\mu$-strongly convex-concave function we have that $V$ is decreasing exponentially as $V(\boldsymbol{\theta}, \boldsymbol{\phi}) = V(\boldsymbol{\theta}_0, \boldsymbol{\phi}_0) \exp(-\mu t)$.*

*Proof.* Similarly as the proof of Theorem 1, we consider the time derivative of $V(\boldsymbol{\theta}, \boldsymbol{\phi})$. For compactness, we are going to simplify the notation slightly by setting $F_i := F_{\boldsymbol{\theta}}(\boldsymbol{x}_i)$, $G_j := G_{\boldsymbol{\phi}}(\boldsymbol{x}'_j)$, $F_i^* := F^*(\boldsymbol{x}_i)$, $G_j^* := G^*(\boldsymbol{x}'_j)$, and $L_{i,j} := L_{\boldsymbol{x}_i, \boldsymbol{x}'_j}$.

$$
\begin{aligned}
\dot{V}(\boldsymbol{\theta}, \boldsymbol{\phi}) &= \frac{1}{n} \sum_{i=1}^n (F_i - F_i^*) \langle \nabla_{\boldsymbol{\theta}} F_i, \dot{\boldsymbol{\theta}} \rangle + \frac{1}{m} \sum_{j=1}^m (G_j - G_j^*) \langle \nabla_{\boldsymbol{\phi}} G_j, \dot{\boldsymbol{\phi}} \rangle \\
&= -\frac{1}{n^2 m} \sum_{i=1}^n (F_i - F_i^*) \langle \nabla_{\boldsymbol{\theta}} F_i, \boldsymbol{A}_{\boldsymbol{\theta}}^\dagger \sum_{(j,k) \in [m] \times [n]} \nabla_{\boldsymbol{\theta}} L_{k,j}(F_k, G_j) \rangle \\
&\quad + \frac{1}{nm^2} \sum_{j=1}^m (G_j - G_j^*) \langle \nabla_{\boldsymbol{\phi}} G_j, \boldsymbol{B}_{\boldsymbol{\phi}}^\dagger \sum_{(k,i) \in [m] \times [n]} \nabla_{\boldsymbol{\phi}} L_{i,k}(F_i, G_k) \rangle \\
&= -\frac{1}{n^2 m} \sum_{i=1}^n (F_i - F_i^*) \sum_{(j,k) \in [m] \times [n]} \langle \nabla_{\boldsymbol{\theta}} F_i, \boldsymbol{A}_{\boldsymbol{\theta}}^\dagger \nabla_{\boldsymbol{\theta}} F_k \rangle \frac{\partial L_{k,j}(F, G_j)}{\partial F} \Big|_{F=F_k} \\
&\quad + \frac{1}{nm^2} \sum_{j=1}^m (G_j - G_j^*) \sum_{(k,i) \in [m] \times [n]} \langle \nabla_{\boldsymbol{\phi}} G_j, \boldsymbol{B}_{\boldsymbol{\phi}}^\dagger \nabla_{\boldsymbol{\phi}} G_k \rangle \frac{\partial L_{i,k}(F_i, G)}{\partial G} \Big|_{G=G_k}.
\end{aligned}
$$

From Lemma 2 we have that $\langle \nabla_{\boldsymbol{\theta}} F_i, \boldsymbol{A}_{\boldsymbol{\theta}}^\dagger \nabla_{\boldsymbol{\theta}} F_k \rangle = n \cdot \delta_{i,k}$, $\forall i \in [n]$, $k \in [m]$, and that $\langle \nabla_{\boldsymbol{\phi}} G_j, \boldsymbol{B}_{\boldsymbol{\phi}}^\dagger \nabla_{\boldsymbol{\phi}} F_k \rangle = m \cdot \delta_{j,k}$. Hence, it follows

$$
\begin{aligned}
\dot{V}(\boldsymbol{\theta}, \boldsymbol{\phi}) &= -\frac{1}{nm} \sum_{\substack{i \in [n] \\ j \in [m]}} (F_i - F_i^*) \frac{\partial L_{i,j}(F, G_j)}{\partial F} \Big|_{F=F_i} + \frac{1}{nm} \sum_{\substack{i \in [n] \\ j \in [m]}} (G_j - G_j^*) \frac{\partial L_{i,j}(F_i, G)}{\partial G} \Big|_{G=G_j} \\
&= -\langle \partial_F L(F, G), F - F^* \rangle + \langle \partial_G L(F, G), G - G^* \rangle
\end{aligned}
$$

where $[\partial_F L(F, G)]_{(i,j)} := \frac{\partial L_{i,j}(F, G_j)}{\partial F} \Big|_{F=F_i}$, $\partial_G L(F, G)]_{(i,j)} := \frac{\partial L_{i,j}(F_i, G)}{\partial G} \Big|_{G=G_j}$, and $F$, $G$, $F^*$, and $G^*$ are indexed by $(i, j)$, as well (while repeating vector elements as necessarily). Thus, using the same reasoning as in Theorem 1, it follows that if $L$ is convex-concave we have that $V$ is non-increasing, and, if $L$ is strictly convex-concave, $\dot{V}(\boldsymbol{\theta}, \boldsymbol{\phi}) < 0$ whenever $(\boldsymbol{\theta}, \boldsymbol{\phi}) \neq (\boldsymbol{\theta}^*, \boldsymbol{\phi}^*)$. Finally, if $L$ is $\mu$-strongly convex-concave, we have by definition that

$$
\begin{aligned}
&-\langle \partial_F L(F, G), F - F^* \rangle + \langle \partial_G L(F, G), G - G^* \rangle \\
&\geq -\mu(\|F - F^*\|^2 + \|G - G^*\|^2) = -\mu V(\boldsymbol{\theta}, \boldsymbol{\phi})
\end{aligned}
$$

Thus we conclude that $V(\boldsymbol{\theta}, \boldsymbol{\phi}) \leq V(\boldsymbol{\theta}_0, \boldsymbol{\phi}_0) \exp(-\mu t)$.

$\square$

### A.4 OMITTED PROOFS OF SECTION 5

In order to prove (17), and (D9) we are, first, going to prove the following useful lemma:

**Lemma 3.** *Let $h_{\boldsymbol{\theta}}(i) := \mathbb{E}_{\mathrm{x}_j \sim \mathrm{Ber}(\theta_j)}[\mathrm{XOR}(\mathrm{x}_1, \ldots, \mathrm{x}_i)]$ where $\boldsymbol{\theta} \in [0, 1]^n$, and $i \leq n$. Then the following equality holds:*

$$
1 - 2h_{\boldsymbol{\theta}}(i) = \prod_{j=1}^i (1 - 2\theta_j). \tag{30}
$$

*Proof.* The easiest way to prove this relationship is by induction on $n \in \mathbb{N}$. For $n = 1$, we only need to verify that (30) holds for $i = 1$. Indeed, we have,

$$1 - 2h_{\boldsymbol{\theta}}(1) = 1 - 2\mathbb{E}_{x \sim \text{Ber}(\theta_1)}[\text{XOR}(x_1)] = 1 - 2\mathbb{E}_{x \sim \text{Ber}(\theta_1)}[x_1] = 1 - 2\theta_1 \,.$$

Next, let us assume that Lemma 3 holds for some $n = n' \in \mathbb{N}$. We are going to prove that Lemma 3 also holds for $n = n' + 1$, and this comes down in proving that (30) holds for $i = n' + 1$:

$$
\begin{aligned}
1 - 2h_{\boldsymbol{\theta}}(n'+1) &= 1 - 2\mathbb{E}[\text{XOR}(x_1, \ldots, x_{n'+1})] = 1 - 2\mathbb{E}[\mathbb{E}[\text{XOR}(x_1, \ldots, x_{n'+1}) \mid x_{n'+1}]] \\
&= 1 - 2(\theta_{n'+1}\mathbb{E}[\text{XOR}(x_1, \ldots, x_{n'}, 1)] + (1 - \theta_{n'+1})\mathbb{E}[\text{XOR}(x_1, \ldots, x_{n'}, 0)]) \\
&= 1 - 2(\theta_{n'+1}\mathbb{E}[1 - \text{XOR}(x_1, \ldots, x_{n'})] + (1 - \theta_{n'+1})\mathbb{E}[\text{XOR}(x_1, \ldots, x_{n'})]) \\
&= 1 - 2(\theta_{n'+1}(1 - h_{\boldsymbol{\theta}}(n')) + (1 - \theta_{n'+1})h_{\boldsymbol{\theta}}(n')) = (1 - 2\theta_{n'+1})(1 - 2h_{\boldsymbol{\theta}}(n')) \\
&= (1 - 2\theta_{n'+1})\prod_{j=1}^{n'}(1 - 2\theta_j) = \prod_{j=1}^{n'+1}(1 - 2\theta_j) \,.
\end{aligned}
$$

And we that, the proof by induction is complete.

$\square$

**Proposition A.4.** *Let* $\mathbf{G} = (n := n_1 + n_2, \mathbb{S} := \{0, 1\}^n, u : \mathbb{S} \to \mathbb{R}^n)$ *be a 2-Team Hidden Matching Pennies game with payoff function given by* (16)*. Then, the expected payoff of the $i$-th member of team $k$ is given by* (17)*.*

*Proof.* The first equality follows, trivially, from the definition of $\Phi(\boldsymbol{s})$. All is left to prove is that $-\mathbb{E}_{s_{k,i} \sim \text{Ber}(\theta_{k,i})}[\Phi(\boldsymbol{s})] = (1 - 2\mathbb{E}_{s_{1,i} \sim \text{Ber}(\theta_{1,i})}[\text{XOR}(\boldsymbol{s}_1)])(1 - 2\mathbb{E}_{s_{2,i} \sim \text{Ber}(\theta_{2,i})}[\text{XOR}(\boldsymbol{s}_2)])$ for all $\boldsymbol{\theta} \in [0, 1]^n$. We define, $x_k = \text{XOR}(\boldsymbol{s}_k)$, $k \in [2]$. Note that, by definition, $x_k \sim \text{Ber}(p_k)$, where $p_k := \mathbb{E}_{s_{k,i} \sim \text{Ber}(\theta_{k,i})}[\text{XOR}(\boldsymbol{s}_k)]$; hence,

$$
\begin{aligned}
\mathbb{E}_{s_{k,i} \sim \text{Ber}(\theta_{k,i})}[\Phi(\boldsymbol{s})] &= \mathbb{E}_{s_{k,i} \sim \text{Ber}(\theta_{k,i})}[1 - 2 \cdot \mathbf{1}_{\text{XOR}(s_1) = \text{XOR}(s_2)}] = \mathbb{E}_{x_k \sim \text{Ber}(p_k)}[1 - 2 \cdot \mathbf{1}_{x_1 = x_2}] \\
&= \mathbb{E}_{x_k \sim \text{Ber}(p_k)}[(1 - 2\,\text{XOR}(x))] = 1 - 2h_{\boldsymbol{p}}(2) = (1 - 2p_1)(1 - 2p_2) \\
&= (1 - 2\mathbb{E}_{s_{1,i} \sim \text{Ber}(\theta_{1,i})}[\text{XOR}(\boldsymbol{s}_1)])(1 - 2\mathbb{E}_{s_{2,i} \sim \text{Ber}(\theta_{2,i})}[\text{XOR}(\boldsymbol{s}_2)]) \,.
\end{aligned}
$$

$\square$

**Proposition A.5.** *Let* $\mathbf{G} = (n := n_1 + n_2, \mathbb{S} := \{0, 1\}^n, u : \mathbb{S} \to \mathbb{R}^n)$ *be a 2-Team Hidden Matching Pennies game with payoff function given by* (16)*. Then, the RD of* $\mathbf{G}$ *is given by the dynamical system of equations:*

$$\dot{\theta}_{k,i} = (-1)^{k-1}\frac{2}{n_k}\theta_{k,i}(1 - \theta_{k,i})\prod_{k' \in [2]}\prod_{\substack{i' \in [n_{k'}] \\ (k,i) \neq (k',i')}}(1 - 2\theta_{k,i}) \,. \tag{D9}$$

*Proof.* From (17) we have that

$$
\begin{aligned}
&\mathbb{E}_{s_{k,i} \sim \text{Ber}(\theta_{k,i})}[u_{k,i}(\boldsymbol{s})] \\
&= (-1)^{k-1}\frac{1}{n_k}(1 - 2\mathbb{E}_{s_{1,i} \sim \text{Ber}(\theta_{1,i})}[\text{XOR}(\boldsymbol{s}_1)])(1 - 2\mathbb{E}_{s_{2,i} \sim \text{Ber}(\theta_{2,i})}[\text{XOR}(\boldsymbol{s}_2)]) \\
&= (-1)^{k-1}\frac{1}{n_k}(1 - 2h_{\boldsymbol{\theta}_1}(n_1)(1 - 2h_{\boldsymbol{\theta}_2}(n_2)) = (-1)^{k-1}\frac{1}{n_k}\prod_{k \in [2]}\prod_{i \in [n_k]}(1 - 2\theta_{k,i}) \,.
\end{aligned}
$$

Then, by the definition of RD of $\mathbf{G}$, we get

$$\dot{\boldsymbol{\theta}}_{k,i} := \theta_{k,i}\mathbb{E}_{s_{k,i}\sim\mathrm{Ber}(\boldsymbol{\theta}_{k,i})}[u_{k,i}(0,\mathbf{s}_{-k,i}) - u_{k,i}(\mathbf{s})]$$

$$= (-1)^{k-1}\frac{1}{n_k}\theta_{k,i}\left(\prod_{k'\in[2]}\prod_{\substack{i'\in[n_{k'}]\\(k,i)\neq(k',i')}}(1-2\theta_{k,i}) - \prod_{k'\in[2]}\prod_{i'\in[n_{k'}]}(1-2\theta_{k,i})\right)$$

$$= (-1)^{k-1}\frac{2}{n_k}\theta_{k,i}(1-\theta_{k,i})\prod_{k'\in[2]}\prod_{\substack{i'\in[n_{k'}]\\(k,i)\neq(k',i')}}(1-2\theta_{k,i})\,.$$

$\square$

**Theorem 3.** *Consider the Replicator Dynamics of* **G**. *Given any interior initial condition, the resulting orbit is a cycle that satisfies the following* $n_1 + n_2 - 1$ *independent invariant functions:*

$$V_{i_1,i_2}(\boldsymbol{\theta}) = \sum_{k=1}^{2} n_k[\log(\theta_{k,i_k}) + \log(1-\theta_{k,i_k})], \; i_k \in [n_k], k \in [2]\,. \tag{19}$$

*Proof.* For all $k \in [2]$ and $i \in [n_k]$, we have

$$\frac{\partial V_{i_1 i_2}(\boldsymbol{\theta})}{\partial\theta_{k,i}} = \begin{cases} \dfrac{n_k(1-2\theta_{k,i})}{\theta_{k,i}(1-\theta_{k,i})}, & \text{if } (k,i) \in \{(k',i_{k'}) \mid k' \in [2]\} \\ \qquad\qquad 0, & \text{otherwise}\,. \end{cases}$$

Subsequently, we have

$$\dot{V}_{i_1 i_2}(\boldsymbol{\theta}) = \langle \nabla_{\boldsymbol{\theta}} V_{i_1 i_2}(\boldsymbol{\theta}), \dot{\boldsymbol{\theta}}\rangle = 2\prod_{k\in[2]}\prod_{i\in[n_k]}(1-2\theta_{k,i}) - 2\prod_{k\in[2]}\prod_{i\in[n_k]}(1-2\theta_{k,i}) = 0\,.$$

Observe that the above imply the existence of $n_1 + n_2 - 1$ independent invariant functions. Hence, the dynamics converge to a limit set of a single degree of freedom, i.e., a cycle.

$\square$

**Proposition A.6.** *Let* **G** $= (n := n_1 + n_2, \mathbb{S} := \{0,1\}^n, u : \mathbb{S} \to \mathbb{R}^n)$ *be a 2-Team Hidden Matching Pennies game with payoff function given by* (16). *Given a mixed-strategy profile* $(\boldsymbol{\theta}, 1 - \boldsymbol{\theta})$, $\boldsymbol{\theta} := (\boldsymbol{\theta}_1, \boldsymbol{\theta}_2) \in (0,1)^n$, $(\boldsymbol{\theta}, 1 - \boldsymbol{\theta})$ *is an equilibrium of* **G** *if and only if* $\exists i_k$, $\forall k \in [2]$ *such that* $\theta_{k,i_k} = \frac{1}{2}$, $\forall k \in [2]$.

*Proof.* We know that **G** is equivalent to a Hidden Matching Pennies game (Example 1) with payoff $\Psi(\boldsymbol{\theta}) = (1 - 2f(\boldsymbol{\theta}_1))(1 - 2g(\boldsymbol{\theta}_2))$, where $f(\boldsymbol{\theta}_1) := \mathbb{E}_{s_{1,i}\sim\mathrm{Ber}(\theta_{1,i})}[\mathrm{XOR}(\mathbf{s}_1)]$, and $g(\boldsymbol{\theta}_2) := \mathbb{E}_{s_{2,i}\sim\mathrm{Ber}(\theta_{2,i})}[\mathrm{XOR}(\mathbf{s}_2)]$ are its hidden mappings. The only fully mixed-Nash equilibrium of this Hidden Matching Pennies game is $(f(\boldsymbol{\theta}_1), g(\boldsymbol{\theta}_2)) = (\frac{1}{2}, \frac{1}{2})$; hence, a mixed-strategy profile $(\boldsymbol{\theta}, 1 - \boldsymbol{\theta})$ is a fully mixed-Nash equilibrium of **G**, if and only if $f(\boldsymbol{\theta}_1) = g(\boldsymbol{\theta}_2) = \frac{1}{2}$. From $f(\boldsymbol{\theta}_1) = \frac{1}{2}$ we get

$$\mathbb{E}_{s_{1,i}\sim\mathrm{Ber}(\theta_{1,i})}[\mathrm{XOR}(\mathbf{s}_1)] = \frac{1}{2} \iff 1 - 2h_{\boldsymbol{\theta}_1}(n_1) = 0 \iff \prod_{i=1}^{n_1}(1-2\theta_{1,i}) = 0$$

$$\iff \exists i_1 \in [n_1] : \theta_{1,i} = \frac{1}{2}\,.$$

$\square$

We remark that, for any $i_k \in [n_k]$, $k \in [2]$,

$$V_{i_1,i_2}(\boldsymbol{\theta}) := \sum_{k=1}^{2} n_k [\log(\theta_{k,i_k}) + \log(1 - \theta_{k,i_k})] = \sum_{k=1}^{2} 2 \cdot n_k [\tfrac{1}{2} \log(\theta_{k,i_k}) + \tfrac{1}{2} \log(1 - \theta_{k,i_k})]$$

$$= \sum_{k=1}^{2} 2 \cdot n_k H(\mathrm{Ber}(\tfrac{1}{2}), \mathrm{Ber}(\theta_{k,i_k}))$$

where $H(p, q)$ is the cross-entropy of a distribution $q$ relative to a distribution $p$. In other words, every one of the invariant functions in (19) is the weighted sum of the cross entropy of two mixed-strategies (of the $i_1$-th member of team 1, and the $i_2$-th member of team 2) relative to the uniform strategy. That is, each invariant function measures (up to a constant) the Kullback–Leibler divergence of two opposing members to an actual equilibrium of the game. Notice that, for any equilibrium point of the Hidden Matching Pennies game, there exists at least on such pair of opposing members $i_1 \in [n_1]$, and $i_2 \in [n_2]$ such that

$$D_{\mathrm{KL}}(\mathrm{Ber}(\tfrac{1}{2}) \parallel \mathrm{Ber}(\theta_{1,i_1})) = D_{\mathrm{KL}}(\mathrm{Ber}(\tfrac{1}{2}) \parallel \mathrm{Ber}(\theta_{2,i_2})) = 0 \,.$$

For the rest of this section we are going to consider the RD of a 2-Team Hidden Matching Pennies game $\mathbf{G} = (n := n_1 + n_2, \mathbb{S} := \{0, 1\}^n, u : \mathbb{S} \to \mathbb{R}^n)$ in a restricted setting, given by the following dynamical system of equations:

$$\dot{\theta}_{k,i} = \begin{cases} 0, & \text{if } \theta_{k,i} \notin \mathbb{S}_{k,i} \\ 0, & \text{if } \theta_{k,i} = \alpha_{k,i} \text{ and } (-1)^k D_{k,i}(\boldsymbol{\theta}) < 0 \\ 0, & \text{if } \theta_{k,i} = \beta_{k,i} \text{ and } (-1)^k D_{k,i}(\boldsymbol{\theta}) > 0 \\ (-1)^{k-1} \dfrac{2}{n_k} \theta_{k,i} (1 - \theta_{k,i}) D_{k,i}(\boldsymbol{\theta}), & \text{otherwise} \end{cases} \qquad \text{(D10)}$$

where $D_{k,i}(\boldsymbol{\theta}) := \prod_{k' \in [2]} \prod_{\substack{j \in [n_{k'}] \\ (k',j) \neq (k,i)}} (1 - 2\theta_{k',j})$, $k \in [2]$ $i \in [n_k]$, and where we restrict each mixed-strategy profile $(\boldsymbol{\theta}, 1 - \boldsymbol{\theta})$ such that $\boldsymbol{\theta}_{k,i} \in \mathbb{S}_{k,i} := [\alpha_{k,i}, \beta_{k,i}]$, $\forall k \in [2]$, $i \in [n_k]$. We let $\Omega$ be an orbit defined by this RD whose initial conditions satisfy Assumption 3. This assumption serves a dual purpose. To begin with, it ensures that any orbit is initialized inside the restricted parameter space that is defined by $\mathbb{S}_{k,i}$. Furthermore, it makes sure the initial strategy profile has full support, i.e., $(\theta_{k,i}(0), 1 - \theta_{k,i}(0))$ is an interior point of the simplex for all $k \in [2]$, and $i \in [n_k]$. It is easy to see that any dimension of the strategy space initially without support is impossible to be updated by the RD in (D10); hence, it would be irrelevant for the analysis. Before we proceed, we are going to introduce a couple of useful lemmas.

**Lemma 4.** $\forall k \in [2]$, and $i \in [n_k]$ if $\theta_{k,i}(0) \in \mathbb{S}_{k,i}$, then $\theta_{k,i}(t) \in \mathbb{S}_{k,i}$, $\forall t \geq 0$.

*Proof.* Let $k \in [2]$ and $i \in [n_k]$ such that $\theta_{k,i}(0) \in \mathbb{S}_{k,i} \implies \alpha_{k,i} \leq \theta_{k,i}(0) \leq \beta_{k,i}$. We are going to prove our case by abduction.

Suppose $\exists t_0 > 0$ such that $\theta_{k,i}(t_0) \notin \mathbb{S}_{k,i}$ and, without any loss of the generality, let us assume that $\theta_{k,i}(t_0) > \beta_{k,i}$. We let $\mathbb{T} = \{t \in (0, t_0) \mid \theta_{k,i}(t) = \beta_{k,i}$ and we note that, since $\theta_{k,i}(0) \leq \beta_{k,i}$, it is implied by the continuity of $\theta_{k,i}(t)$ (Equation D10) and by the *Intermediate Value Theorem* that $\mathbb{T} \neq \emptyset$. Finally we define $t_{\max} = \max(\mathbb{T})$.

Let us now consider the value of $\theta_{k,i}(t)$ for some $t \in (t_{\max}, t_0)$ and observe that if $\theta_{k,i}(t) = \beta_{k,i}$ we have

$$t \in \mathbb{T} \implies t \leq \max(\mathbb{T}) = t_{\max} \implies t_{\max} < t \leq t_{\max} \,.$$

This contradiction implies that $\theta_{k,i}(t) \neq \beta_{k,i}$. However, if $\theta_{k,i}(t) < \beta_{k,i}$, the Intermediate Value Theorem, once again, implies $\exists t' \in (t, t_0)$ such that $\theta_{k,i}(t') = \beta_{k,i}$, which as before implies $t_{\max} < t' \leq t_{\max}$. Hence, it must be the case that, for all $t \in (t_{\max}, t_0)$:

$$\theta_{k,i}(t) > \beta_{k,i} \overset{(D10)}{\Longrightarrow} \dot{\theta}_{k,i}(t) = 0\,.$$

However, by applying the *Mean Value Theorem*, if follows $\exists t' \in (t_{\max}, t_0)$ such that

$$\dot{\theta}_{k,i}(t) = \frac{\theta_{k,i}(0) - \theta_{k,i}(t_{\max})}{t_0 - t_{\max}} = \frac{\theta_{k,i}(0) - \beta_{k,i}}{t_0 - t_{\max}} > 0\,,$$

which is once again a contradiction.

$\square$

**Lemma 5.** *Under Assumption 3, if $\mathbb{S}_k = \emptyset$, $\forall k \in [2]$ then the following hold:*

(a) $\Psi_{1,i_1}(\boldsymbol{\theta}(t))\Psi_{2,i_2}(\boldsymbol{\theta}(t)) < 0$ *for all $i_k \in [n_k]$, $k \in [2]$, and $t \geq 0$.*

(b) $\Psi_{k,i_k}(\boldsymbol{\theta}(t))$ *preserves sign for all $i_k \in [n_k]$, $k \in [2]$.*

*Proof.* Since $\mathbb{S}_k = \emptyset$ for all $k \in [2]$, it follows, by definition, that

$$\frac{1}{2} \notin \mathbb{S}_{k,i}, \ \forall k \in [2], \ i \in [n_k]$$

Furthermore, by Assumption 3, we have that $\boldsymbol{\theta}_{k,i}(0) \in \mathbb{S}_{k,i}$ for all $k \in [2]$, $i \in [n_k]$. Hence, it follows, by Lemma 4, that $\forall k \in [2]$, $i \in [n_k]$, and $t \geq 0$:

$$\boldsymbol{\theta}_{k,i}(t) \in \mathbb{S}_{k,i} \implies \boldsymbol{\theta}_{k,i}(t) \neq \frac{1}{2} \implies 1 - 2\boldsymbol{\theta}_{k,i}(t) \neq 0\,.$$

Then, by Equation 17, we have

$$\begin{aligned}
\Psi_{k,i}(\boldsymbol{\theta}(t)) &= \frac{(-1)^{k-1}}{n_k}(1 - 2\mathbb{E}_{\mathbf{s}_{1,j} \sim \mathrm{Ber}(\theta_{1,j})}[\mathrm{XOR}(\mathbf{s}_1)])(1 - 2\mathbb{E}_{\mathbf{s}_{2,j} \sim \mathrm{Ber}(\theta_{2,j})}[\mathrm{XOR}(\mathbf{s}_2)]) \\
&= \frac{(-1)^{k-1}}{n_k}(1 - 2h_{\boldsymbol{\theta}_1})(1 - 2h_{\boldsymbol{\theta}_2}) = \frac{(-1)^{k-1}}{n_k}\prod_{j=1}^{n_1}(1 - 2\theta_{1,j})\prod_{j=1}^{n_2}(1 - 2\theta_{2,j}) \\
&= \frac{(-1)^{k-1}}{n_k}\prod_{k'=1}^{2}\prod_{j=1}^{n_{k'}}(1 - 2\theta_{k',j}(t)) \neq 0\,.
\end{aligned}$$

That implies (a). We can now condition on $\Psi_{k,i}(\boldsymbol{\theta}(0))$. Since $\Psi_{k,i}(\boldsymbol{\theta}(t)) \neq 0$ for all $t \geq 0$, it must be the case that either $\Psi_{k,i}(\boldsymbol{\theta}(0)) > 0$ or $\Psi_{k,i}(\boldsymbol{\theta}(0)) < 0$; let us, first, assume the former case. We are going to prove, by abduction, that $\Psi_{k,i}(\boldsymbol{\theta}(t)) > 0$, $\forall t \geq 0$.

Suppose $\exists t' > 0$ such that $\Psi_{k,i}(\theta(t')) \leq 0$. Since $\Psi_{k,i}(\boldsymbol{\theta}(t)) \neq 0$ for all $t \geq 0$ it follows that $\Psi_{k,i}(\boldsymbol{\theta}(t')) < 0$ must be the case. However, by the continuity of $\Psi_{k,i}(\boldsymbol{\theta}(t))$ and the Intermediate Value Theorem, it follows that $\exists t'' \in (0, t')$ such that $\Psi_{k,i}(\boldsymbol{\theta}(t'')) = 0$, and that is, indeed, a contradiction. It follows that it must be the case $\Psi_{k,i}(\boldsymbol{\theta}(t)) > 0$, $\forall t \geq 0$.

In a similar manner we may prove that $\Psi_{k,i}(\boldsymbol{\theta}(0)) < 0 \implies \Psi_{k,i}(\boldsymbol{\theta}(t)) < 0$, $\forall t \geq 0$; hence, (b) holds.

$\square$

**Theorem 4.** *Under Assumption 3, if $\mathbb{S}_k = \emptyset$, $\forall k \in [2]$, the restricted RD of **G** converge to*

$$
\theta_{1,i}^* = \begin{cases} \alpha_{1,i}, & \text{if } \beta_{1,i} < \frac{1}{2} \\ \beta_{1,i}, & \text{otherwise} \end{cases} \quad i \in [n_1] \qquad \theta_{1,i}^* = \begin{cases} \beta_{1,i}, & \text{if } \beta_{1,i} < \frac{1}{2} \\ \alpha_{1,i}, & \text{otherwise} \end{cases} \quad i \in [n_1]
$$

$$
\underbrace{\theta_{2,i}^* = \begin{cases} \beta_{2,i}, & \text{if } \beta_{2,i} < \frac{1}{2} \\ \alpha_{2,i}, & \text{otherwise} \end{cases} \quad i \in [n_2]}_{\text{if } |\mathbb{S}| \text{ is even}} \quad or \quad \underbrace{\theta_{2,i}^* = \begin{cases} \alpha_{2,i}, & \text{if } \beta_{2,i} < \frac{1}{2} \\ \beta_{2,i}, & \text{otherwise} \end{cases} \quad i \in [n_2]}_{\text{if } |\mathbb{S}| \text{ is odd}} \qquad (20)
$$

*where $\mathbb{S} := \{(k,i) \mid k \in [2], i \in [n_k], \alpha_{k,i} > \frac{1}{2}\}$.*

*Proof.* We begin by conditioning on the value of $|\mathbb{S}|$ and, since the proof is similar in both cases, and without any loss of the generality, we are going to assume that $|\mathbb{S}|$ is even. Since $\mathbb{S}_k = \emptyset$ for all $k \in [2]$, i.e., $\frac{1}{2} \notin \mathbb{S}_{k,i}$ for all $k \in [2]$, $i \in [n_k]$, and, by Assumption 3, $\theta_{k,i}(0) \in \mathbb{S}_{k,i} := [\alpha_{k,i}, \beta_{k,i}]$ for $k \in [2]$, $i \in [n_k]$ it follows that $\theta_{k,i}(0) > \frac{1}{2}$, if $\alpha_{k,i} > \frac{1}{2}$; $\theta_{k,i}(0) < \frac{1}{2}$; otherwise. Then, for all $i \in [n_1]$, we have

$$
\Psi_{1,i}(\boldsymbol{\theta}(0)) = \frac{1}{n_k} \prod_{k=1}^{2} \prod_{j=1}^{n_k} (1 - 2\theta_{k,j}(0))
$$

$$
= \frac{1}{n_k} \left( \prod_{(k,j) \in \mathbb{S}} (1 - 2\theta_{k,j}(0)) \right) \cdot \left( \prod_{(k,j) \notin \mathbb{S}} (1 - 2\theta_{k,j}(0)) \right) > 0,
$$

since $|\mathbb{S}|$ is even. Subsequently, by Lemma 5, $\Psi_{1,i}(\boldsymbol{\theta}(t)) > 0$, $\forall t \geq 0$. That implies that $\Psi_{1,i}(\boldsymbol{\theta}(t)) > 0$, $\forall i \in [n_1]$, $t \geq 0$, and, hence, by Lemma 5, we also have that $\Psi_{2,i}(\boldsymbol{\theta}(t)) < 0$, $\forall i \in [n_2]$, $t \geq 0$. Take any $k \in [2]$, and $i \in [n_k]$. In order to complete the proof, we'll have to condition on the value of $k$, and on whether $(k,i) \in \mathbb{S}$. There are four cases in total that we"ll have to consider, but, since in all of them we follow a similar reasoning, we'll just go ahead and demonstrate the single case of $k = 1$, and $(k,i) \in \mathbb{S}$.

We are going to prove that there exists $t_{1,i}^* \geq 0$ such that $\theta_{1,i}(t) = \beta 1, i$, $\forall t \geq t_{1,i}^*$. By Assumption 3, and Lemma 4, we have $\alpha_{1,i} \geq \theta_{1,i}(t) \geq \beta_{1,i}$ for all $t \geq 0$. Furthermore, since $(1,i) \in \mathbb{S}$, and, since $\mathbb{S}_k = \emptyset$, we have that $\alpha_{1,i} \geq \beta_{1,i} > \frac{1}{2}$. That is, $\theta_{1,i}(t) > \frac{1}{2}$ for all $t \geq 0$, and, hence,

$$
D_{1,i}(\boldsymbol{\theta}(t)) := \prod_{k \in [2]} \prod_{\substack{j \in [n_k] \\ (k,j) \neq (1,i)}} (1 - 2\theta_{k,j}(t)) = \frac{1}{1 - 2\theta_{1,i}(t)} \cdot \prod_{k=1}^{2} \prod_{j=1}^{n_k} (1 - 2\theta_{k,j}(t))
$$

$$
= n_1 \cdot \frac{1}{1 - 2\theta_{1,i}(t)} \cdot \Psi_{1,i}(t) < 0.
$$

Then, by Equation D10 it follows that $\forall t \geq 0$:

$$
\dot{\theta}_{1,i}(t) = \begin{cases} 0, & \text{if } \theta_{1,i} = \beta_{1,i} \\ \frac{2}{n_1} \theta_{1,i}(1 - \theta_{1,i}) D_{1,i}(\boldsymbol{\theta}), & \text{otherwise}. \end{cases} \qquad (D11)
$$

Hence, $\theta_{1,i} = \beta_{1,i}$ is the single attracting point of the dynamical system described by D11 and, hence, by definition, $\exists t_{1,i}^* \geq 0$ such that $\theta_{1,i}(t) = \theta_{1,i}^* = \beta_{1,i}$, $\forall t \geq t_{1,i}^*$. Similarly, we can prove that $\exists t_{k,i}^* \geq 0$ such that $\theta_{k,i}(t) = \theta_{k,i}^*$, $\forall t \geq t_{k,i}^*$, and by letting $t^* = \max_{\substack{k \in [2] \\ i \in [n_k]}} (t_{k,i}^*)$, Theorem 4 follows.

$\square$

**Theorem 5.** *Under Assumption 3, if $\mathbb{S}_k \neq \emptyset$, $\forall k \in [2]$, then the restricted RD of $\mathbf{G}$ converge to an invariant set defined by the $|\mathbb{S}_1| + |\mathbb{S}_2| - 1$ independent invariant functions, $V_{i_1,i_2}(\boldsymbol{\theta})$, $i_k \in \mathbb{S}_k$, $\forall k \in [2]$, where $V_{i_1,i_2}(\boldsymbol{\theta})$ is given as in (19).*

*Proof.* We are going to begin by defining the following time-dependent set:

$$
\mathbb{C}(t) := \begin{cases} \{(k,i) \mid k \in [2],\, i \in [n_k],\, \theta_{k,i}(t) \in \{\alpha_{k,i}, \beta_{k,i}\}\}, & \text{if } t \geq 0 \\ \emptyset, & \text{otherwise}. \end{cases}
$$

We perform a time partitioning based on the values of $\mathbb{C}(t)$, $t \geq 0$. Specifically, we let $\mathbb{P} = \{(t_1,t_2) \mid t_1, t_2 \in \mathbb{T} : t \notin \mathbb{T},\ \forall t \in (t_1, t_2)\}$, where

$$
T := \{t \geq 0 \mid \exists \epsilon_0 > 0 : \mathbb{C}(t - \epsilon) \neq \mathbb{C}(t),\ \forall \epsilon \in (0, \epsilon_0)\}.
$$

It is not difficult to see that the continuity of $\boldsymbol{\theta}(t)$ implies that $\mathbb{T}$ consists entirely of isolated points and, due to this fact, $\mathbb{P}$ is well-defined. Let $\mathbb{I} \in \mathbb{P}$ be one of these partitions. We are going to prove that $\dot{V}_{i_1,i_2}(\boldsymbol{\theta}(t)) \geq 0$, $\forall i_k \in \mathbb{S}_k$, $\forall k \in [2]$, $t \in \mathbb{I}$.

First of all, observe that, by definition, the following two properties have to hold:

(a) $\mathbb{C}(t_1) = \mathbb{C}(t_2)$, $\forall t_1, t_2 \in I$

(b) By Equation D10, we have that $\forall t \in I$,

$$
\dot{\theta}_{k,i}(t) = \begin{cases} 0, & \text{if } (k,i) \in \mathbb{C}(t) \\ (-1)^{k-1} \dfrac{2}{n_k} \theta_{k,i}(1 - \theta_{k,i}) D_{k,i}(\boldsymbol{\theta}), & \text{otherwise}. \end{cases} \tag{D12}
$$

For any $i_k \in \mathbb{S}_k$, $k \in [2]$, we proceed by conditioning on the value of $\mathbb{C}(t)$, $t \in I$ and, since $\mathbb{C}(t_1) = \mathbb{C}(t_2)$, $\forall t_1, t_2 \in I$, we distinct only three cases:

(a) $(k, i_k) \in \mathbb{C}(t)$, $\forall k \in [2]$, $t \in \mathbb{I}$.

(b) $(k, i_k) \notin \mathbb{C}(t)$, $\forall k \in [2]$, $t \in \mathbb{I}$.

(c) $\exists k, k' \in [2]$ such that $(k, i_k) \in \mathbb{C}(t)$ and $(k', i_{k'}) \notin \mathbb{C}(t)$ for all $t \in \mathbb{I}$.

The analysis of the first two cases is relatively straightforward. Let $t \in \mathbb{I}$; then, if $(k, i_k) \in \mathbb{C}(t)$, $\forall k \in [2]$ is the case, then by Equation D12, we have that $\dot{\theta}_{k,i_k}(t) = 0$, $\forall k \in [2]$ and, hence,

$$
\dot{V}_{i_1,i_2}(\boldsymbol{\theta}(t)) = (\nabla_{\boldsymbol{\theta}(t)} V_{i_1,i_2}(\boldsymbol{\theta}(t)))^{\mathsf{T}} \dot{\boldsymbol{\theta}}(t) = \sum_{k=1}^{2} \left( \frac{n_k(1 - 2\theta_{k,i}(t))}{\theta_{k,i}(t)(1 - \theta_{k,i}(t))} \cdot \dot{\theta}_{k,i_k}(t) \right) = 0.
$$

On the other hand, if $(k, i_k) \notin \mathbb{C}(t)$, $\forall k \in [2]$ is the case, then, once again, by Equation D12, we have that $\dot{\theta}_{k,i_k}(t) = (-1)^{k-1} \frac{2}{n_k} \theta_{k,i}(1 - \theta_{k,i}) D_{k,i}(\boldsymbol{\theta})$, $\forall k \in [2]$. That implies,

$$
\dot{V}_{i_1,i_2}(\boldsymbol{\theta}(t)) = \sum_{k=1}^{2} \left( \frac{n_k(1 - 2\theta_{k,i}(t))}{\theta_{k,i}(t)(1 - \theta_{k,i}(t))} \cdot (-1)^{k-1} \frac{2}{n_k} \theta_{k,i}(t)(1 - \theta_{k,i}(t)) D_{k,i}(\boldsymbol{\theta}(t)) \right)
$$

$$
= 2 \cdot \prod_{k=1}^{2} \prod_{j=1}^{n_k} (1 - 2\theta_{k,j}(t)) \cdot \sum_{k=1}^{2} (-1)^{k-1} = 0.
$$

In order to proceed with the final case we'll first need to prove a small technical lemma:

**Lemma 6.** *If $(k, i) \in \mathbb{C}(t)$ for some $t \in \mathbb{I}$, $\mathbb{I} \in \mathbb{P} = \{(t_1, t_2) \mid t_1, t_2 \in \mathbb{T} : t \notin \mathbb{T}, \forall t \in (t_1, t_2)\}$ then one of the following holds:*

*a) $\theta_{k,i}(t) = \alpha_{k,i}$, $\forall t \in \mathbb{I}$.*

*b) $\theta_{k,i}(t) = \beta_{k,i}$, $\forall t \in \mathbb{I}$.*

To see that, let $\mathbb{I} \in \mathbb{P}$ and $(k, i) \in \mathbb{C}(t)$ for some $t \in \mathbb{I}$. Then it, holds, by definition, that

$$\mathbb{C}(t_1) = \mathbb{C}(t_2), \ \forall t_1, t_2 \in \mathbb{I} \implies \mathbb{C}(t') = \mathbb{C}(t), \ \forall t' \in \mathbb{I}$$

And, since $(k, i) \in \mathbb{C}(t)$, we also have that

$$(k, i) \in \mathbb{C}(t'), \ \forall t' \in \mathbb{I} \implies \theta_{k,i}(t') \in \{\alpha_{k,i}, \beta_{k,i}\}$$

Let us assume that $\theta_{k,i}(t) = \alpha_{k,i}$ and suppose $\exists t' \in \mathbb{I}$ such that $\theta_{k,i}(t') \neq \alpha_{k,i} \implies \theta_{k,i}(t') = \beta_{k,i}$ and $\alpha_{k,i} \neq \beta_{k,i}$. By the Intermediate Value Theorem, it follows $\exists t'' \in (\min(t, t'), \max(t, t'))$ such that

$$\theta_{k,i}(t'') \in (\alpha_{k,i}, \beta_{k,i}) \implies (k, i) \notin \mathbb{C}(t'')$$

That is a contradiction, and, hence, a) holds. a) follows by a similar argument, assuming $\theta_{k,i}(t) = \beta k, i$ is the case, instead.

Having established Lemma 6, we continue with the proof of Theorem 5. Let us assume that $\exists k, k' \in [2]$ such that $(k, i_k) \in \mathbb{C}(t)$ and $(k', i_{k'}) \notin \mathbb{C}(t)$. Then Lemma 6 implies that either $\theta_{k,i_k}(t') = \alpha_{k,i_k}$, $\forall t' \in \mathbb{I}$ or $\theta_{k,i_k}(t') = \beta_{k,i_k}$, $\forall t' \in \mathbb{I}$. Let us assume, without any loss of the generality, the former case and let us consider some $t \in \mathbb{I}$. Since $\mathbb{S}_{k,i_k} \in \mathbb{S}_k$, i.e., $\alpha_{k,i_k} \leq \frac{1}{2}$, that implies $1 - 2\theta_{k,i_k}(t) \geq 0$. Next, Let us consider the value of $(-1)^{k-1} D_{k,i}(\boldsymbol{\theta})$. We are going to prove by abduction that $(-1)^{k-1} D_{k,i}(\boldsymbol{\theta}) \leq 0$.

If $(-1)^{k-1} D_{k,i}(\boldsymbol{\theta}) > 0$ then by Equation D10 we have that$\dot{\theta}_{k,i_k}(t) = (-1)^{k-1} \frac{2}{n_k} \theta_{k,i}(1 - \theta_{k,i}) D_{k,i}(\boldsymbol{\theta}) \neq 0$, where the last inequality follows by Assumption 3. By the continuity $\boldsymbol{\theta}(t)$, it follows that $\exists \epsilon_0 > 0$ such that $\forall \epsilon \in (0, \epsilon_0)$, $\theta_{k,i_k}(t + \epsilon) \neq \theta_{k,i_k}(t) = \alpha_{k,i_k} \implies (k, i_k) \notin \mathbb{C}(t)$; that is, indeed, a contradiction. It must then be the case that $(-1)^{k-1} D_{k,i}(\boldsymbol{\theta}) \leq 0$ and, thus, we have that

$$\dot{V}_{i_1,i_2}(\boldsymbol{\theta}(t)) = \frac{n_{k'}(1 - 2\theta_{k',i}(t))}{\theta_{k',i}(t)(1 - \theta_{k',i}(t))} \cdot (-1)^{k'-1} \frac{2}{n_{k'}} \theta_{k',i}(t)(1 - \theta_{k',i}(t)) D_{k',i}(\boldsymbol{\theta}(t))$$

$$= 2 \cdot (-1)^{k'-1} \cdot (1 - 2\theta_{k',i}(t)) D_{k',i}(\boldsymbol{\theta}(t)) = 2 \cdot (-1) \cdot (-1)^{k-1} \cdot (1 - 2\theta_{k,i}(t)) D_{k,i}(\boldsymbol{\theta}(t)) \geq 0.$$

Thus, we showed that in every case $\dot{V}_{i_1,i_2}(\boldsymbol{\theta}(t)) \geq 0$, $\forall t \in \mathbb{I}$. However, since $\boldsymbol{\theta}(t)$ is continuous (since it is differentiable), we can extend this property for any $t \geq 0$, i.e., $\dot{V}_{i_1,i_2}(\boldsymbol{\theta}(t)) \geq 0$, $\forall t \geq 0$. Finally, notice that $V_{i_1,i_2}(\boldsymbol{\theta}(t))$ is bounded, and, hence, it follows that $-V_{i_1,i_2}(\boldsymbol{\theta}(t))$ is a Lyapunov function (up to a constant) of the dynamical system described by Equation D10, and, Theorem 5 follows by the definition of a Lyapunov function.

$\square$

# B SUPPLEMENTARY EXPERIMENTAL RESULTS

## B.1 EXPERIMENTAL RESULTS FOR 2-TEAM HIDDEN MATCHING PENNIES GAMES

In this section we present exemplary settings for the dynamics presented in section 5. For an overview of the conditions that characterize the behaviors presented in these examples, we refer-

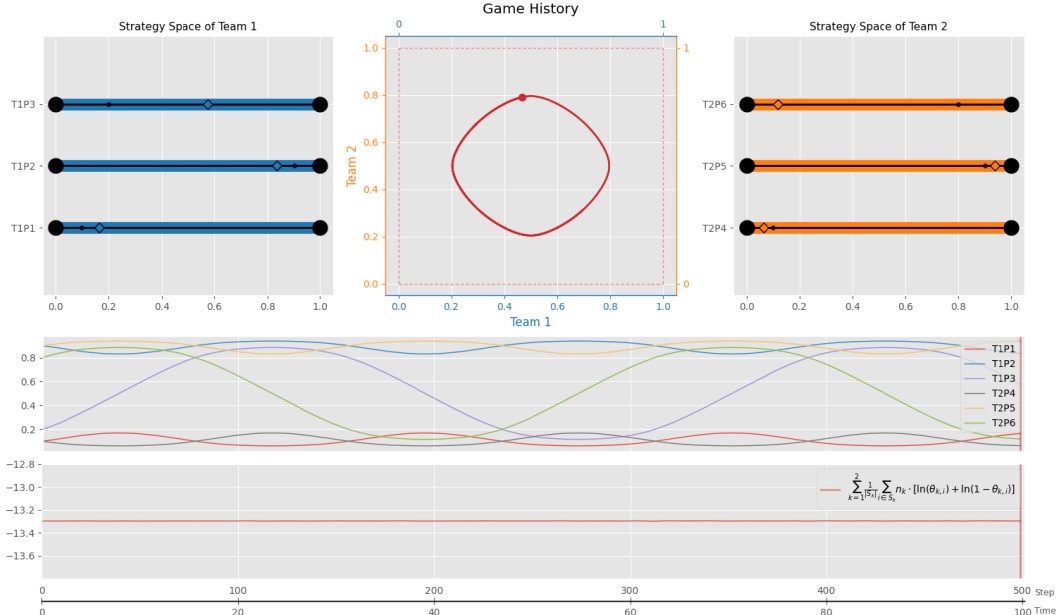

Figure 3: An example of the RD of a 2-Team Hidden Matching Pennies *(Center)* with no restrictions applied in the parameters space $(\boldsymbol{\theta}, \mathbf{1} - \boldsymbol{\theta})$ in *(Left)*, and *(Right)*. In this case, the RD cycles for any initial point $(\boldsymbol{\theta}(0), \mathbf{1} - \boldsymbol{\theta}(0))$. The evolution of $\boldsymbol{\theta}(t)$, as a function of time, $t \in [0, 100]$ is depicted in *(Bottom)* along with the Lyapunov function given in (31).

ence the interested reader to section 5; for formal definitions and the proofs of these concepts, see §A.4.

Let $\mathbf{G} = (n := n_1 + n_2, \mathbb{S} := \{0,1\}^n, u : \mathbb{S} \to \mathbb{R}^n)$ be a 2-Team Hidden Matching Pennies game with $n_1 = n_2 = 3$ and payoff function given by (16). Figure 3 depicts the behavior of the RD in an unrestricted instance of $\mathbf{G}$, i.e., $\mathbb{S}_{k,i} = [0,1]$ for all $i \in [3]$, $k \in [2]$. The restrictions applied to team 1 and team 2 are visualized by the feasible range of each member's strategies (Figure 3 *(Left)*, and *(Right)*, respectively). The black dot in each range indicates the initial strategy of each member, i.e., $\theta_{k,i}(0)$. Note that, by Assumption 3, this strategy profile, $(\boldsymbol{\theta}(0), \mathbf{1} - \boldsymbol{\theta}(0))$, is assumed to lie inside the feasible region enforced by the constraints $\mathbb{S}_{k,i}$.

We solve the initial value problem of the ODE that corresponds to this RD using the RADAU integration method implemented by the *scipy* package in Python 3, and we perform 500 evaluations over the time interval $t \in [0, 100]$. The orbit of the RD with initial parametrization $\boldsymbol{\theta}(0)$ is depicted as a curve on the $\mathbb{F} \times \mathbb{G}$ space defined by the hidden mappings in (18) (Figure 3 *(Center)*). The red dot indicates the the point $(f(\boldsymbol{\theta}_1(t)), g(\boldsymbol{\theta}_2(t))$ at the end of the simulation, i.e., at time $t = 100$. The corresponding strategies of each member are indicated by blue (team 1) and orange (team 2) dots inside the corresponding feasible regions (Figure 3 *(Left)*, and *(Right)*). The individual trajectory of each $\theta_{k,i}$, $i \in [3]$, $k \in [2]$ is depicted in Figure 3 *(Bottom)*, where the curve labeled $\mathrm{T}k\mathrm{P}i$ corresponds to the trajectory of the $i$-th member of team $k$ if $k = 1$, or the $i - n_1$-th member of team $k$ if $k = 2$. In the case that the RD cycle, e.g., in the unrestricted setting, we depict one of the Lyapunov functions of the RD (Figure 3 *(Bottom)*). Specifically, the Lyapunov function of our choice is

$$V(t) := \sum_{k=1}^{2} \frac{1}{|\mathbb{S}_k|} \sum_{i \in \mathbb{S}_k} n_k \cdot [\log(\theta_{k,i}(t)) + \log(1 - \theta_{k,i}(t))], \qquad (31)$$

the average value of all the linearly independent Lyapunov functions defined in (19). We selected this specific Lyapunov function as the average is, in general, more numerically stable and for compactness.

In Figure 3, we verify that the RD of **G**, indeed, cycle in an unrestricted setting (Theorem 3). The choice of initial points does not matter in this case; for completeness, we note that $\boldsymbol{\theta}_1 = (0.1, 0.9, 0.2)$, and $\boldsymbol{\theta}_2 = (0.1, 0.9, 0.8)$ in all of the examples in this section.

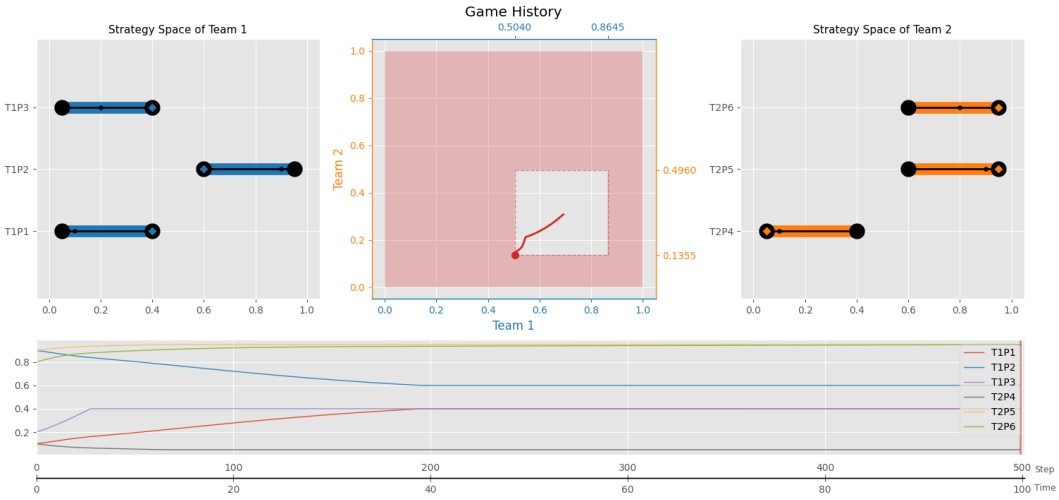

Figure 4: An example of the RD of a 2-Team Hidden Matching Pennies *(Center)* in a restricted setting where $\frac{1}{2} \notin \mathbb{S}_{k,i}$ for all $i \in [3]$, $k \in [2]$ *((Left)*, and *(Right))*. In this case, the RD converges for any initial interior point $(\boldsymbol{\theta}(0), \mathbf{1} - \boldsymbol{\theta}(0))$. The evolution of $\boldsymbol{\theta}(t)$, as a function of time, $t \in [0, 100]$ is depicted in *(Bottom)*.

As the first example in a restricted setting (Figure 4), we are going to enforce $\mathbb{S}_{1,1} = \mathbb{S}_{1,3} = \mathbb{S}_{2,1} = [0.05, 0.4]$, and $\mathbb{S}_{1,2} = \mathbb{S}_{2,2} = \mathbb{S}_{2,3} = [0.6, 0.95]$. First, observe that $\mathbb{S}_{k,i} \subset (0, 1)$ for all $i \in [3]$, $k \in [2]$; hence, Assumption 3 is satisfied. Next, notice that $\frac{1}{2} \notin \mathbb{S}_{k,i}$ for all $i \in [3]$, $k \in [2]$. It follows, by Theorem 4, that the RD should converge in this case (Figure 4 *(Center)*, and *(Bottom)*), and the point of convergence is, indeed, $\boldsymbol{\theta}_1^* = (0.4, 0.6, 0.4)$, and $\boldsymbol{\theta}_2^* = (0.05, 0.95, 0.95)$, as given in Theorem 4.

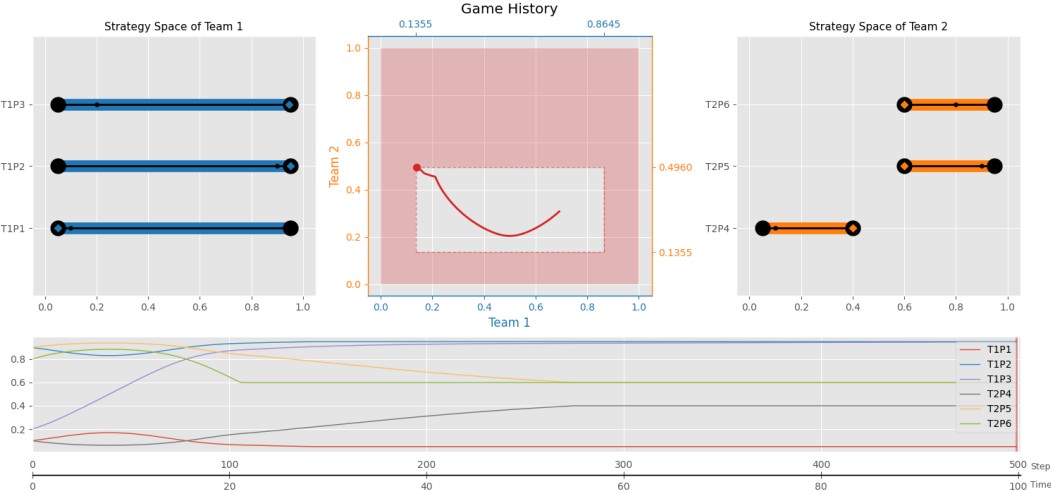

Figure 5: An example of the RD of a 2-Team Hidden Matching Pennies *(Center)* in a restricted setting where $\frac{1}{2} \in \mathbb{S}_{1,i}$, and $\frac{1}{2} \notin \mathbb{S}_{2,i}$ for all $i \in [3]$ *((Left)*, and *(Right))*. In this case the RD converges for any initial interior point $(\boldsymbol{\theta}(0), \mathbf{1} - \boldsymbol{\theta}(0))$, but the point of convergence depends on the value of $\boldsymbol{\theta}(0)$. The evolution of $\boldsymbol{\theta}(t)$, as a function of time, $t \in [0, 100]$ is depicted in *(Bottom)*.

There exists a similar setting (Figure 5), where the RD of **G** converge, as well. Note, in this case, while $\frac{1}{2} \notin \mathbb{S}_{2,i}$ for all $i \in [3]$, we let $\frac{1}{2} \in \mathbb{S}_{1,i}$ for some $i \in [3]$ (specifically, $\frac{1}{2} \in \mathbb{S}_{1,i}$ for all $i \in [3]$ in this particular example). Although we did not provide a convergence analysis for this case, it is not difficult to see that the convergence guarantees from Theorem 4 can be generalized to include this case. As is visualized in Figure 5 *(Center)*, the exhibited behavior is a hybrid of the two behaviors described by Theorem 4, and Theorem 5. In particular, the behavior of the dynamics depends on the quadrant of the $\mathbb{F} \times \mathbb{G}$ space that the $\boldsymbol{\theta}(t)$ lies. Ultimately though, the dynamics can be shown to converge to a point $\boldsymbol{\theta}^*$, which depends not only on the initialization $\boldsymbol{\theta}(0)$ but also on the restrictions in the space parameters, $\mathbb{S}_{k,i}$, $i \in [3]$, $k \in [2]$. The particular point of convergence for this example is $\boldsymbol{\theta}_1^* = (0.05, 0.95, 0.95)$, and $\boldsymbol{\theta}_2^* = (0.4, 0.6, 0.6)$, where the space restrictions are given as $\mathbb{S}_{2,1} = [0.05, 0.4]$, $\mathbb{S}_{2,2} = \mathbb{S}_{2,3} = [0.6, 0.95]$, and $\mathbb{S}_{1,i} = [0.05, 0.95]$ for all $i \in [3]$.

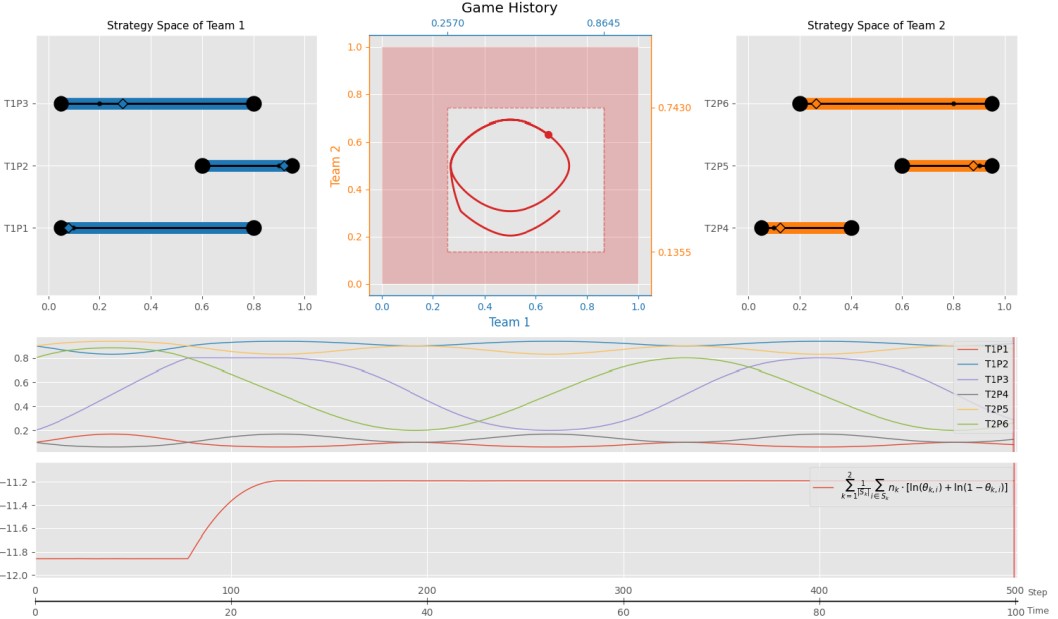

Figure 6: An example of the RD of a 2-Team Hidden Matching Pennies *(Center)* in a restricted setting where $\frac{1}{2} \in \mathbb{S}_{1,i}$, and $\frac{1}{2} \in \mathbb{S}_{2,j}$ for some $i, j \in [3]$ (*(Left)*, and *(Right)*). In this case the RD converge to a cycle defined by, at least, $|\mathbb{S}_1| + |\mathbb{S}_2| - 1$ linearly independent invariant functions for any initial interior point $(\boldsymbol{\theta}(0), \mathbf{1} - \boldsymbol{\theta}(0))$ The evolution of $\boldsymbol{\theta}(t)$, as a function of time, $t \in [0, 100]$ is depicted in *(Bottom)*.

Finally, we present an example of cycling behavior in the restricted setting (Figure 6) where we applied the restrictions $\mathbb{S}_{1,1} = \mathbb{S}_{1,3} = [0.05, 0.8]$, $\mathbb{S}_{1,2} = \mathbb{S}_{2,2} = [0.6, 0.95]$, $\mathbb{S}_{2,1} = [0.05, 0.4]$, and $\mathbb{S}_{2,3} = [0.2, 0.95]$. Note that $\frac{1}{2} \in \mathbb{S}_{1,1}, \mathbb{S}_{1,3}, \mathbb{S}_{2,3}$; hence, $|\mathbb{S}_1| + |\mathbb{S}_2| = 3$, and there exists $i_k \in [n_k]$ for all $k \in [2]$ such that $\frac{1}{2} \in \mathbb{S}_{k,i_k}$. By Theorem 5, it follows that the restricted RD cycle in this case; specifically, they converge to an invariant set defined by $V_{1,3}(\boldsymbol{\theta}) = 0$, and $V_{3,3}(\boldsymbol{\theta}) = 0$ as given by (19). This behavior is depicted in Figure 6 *(Center)*. Notice how the Lyapunov function, $V(\boldsymbol{\theta})$ (Figure 6 *(Bottom)*) is monotonically increasing, and stabilizes at approximately $t = 25$.

