# OpenReview forum: "Generalized Natural Gradient Flows in Hidden Convex-Concave Games and GANs"
_ICLR.cc/2022/Conference — ICLR 2022 Poster_

### Official Review · Reviewer_qQLu · 2021-10-29

**Correctness:** 3
**Technical Novelty And Significance:** 3
**Empirical Novelty And Significance:** Not applicable
**Recommendation:** 6
**Confidence:** 2

**Main Review:**

The ideas driving this paper are interesting, but the mathematical rigor is somewhat questionable. There are several points, partially detailed below, where either the presentation is unclear because of many flaws and typos in the writing, or at least not rigorously established.

Furthermore, the analysis section is rather limited in terms of novelty. Proposition 1, can be deduced from Vlatakis-Gkaragkounis, E. V., Flokas, L., & Piliouras, G. (2020). Solving Min-Max Optimization with Hidden Structure via Gradient Descent Ascent.

The strong and interesting parts of the paper are in Section 4.2 where a Lyapunov function for the dynamics on a finite sample is constructed. To form of the Lyapunov function is as expected, but the analysis is quite nice.

The weak parts are the sometimes ambiguous and unclear notation and a serious discussion on the computational issues involved in the proposed approach. In particular, there a several points the authors should take care of:

There are several technical problems with the paper.

	•	I think the notation $\varphi$ (for the saddle function) and $\phi$ for the parameter of the generator is very unfortunate. Also, the notation for the domains of the mapping $F$ and $G$ seems to me messed up all over the paper. Be consistent with your notation.

	•	The way how example 2 is presented in misleading in my opinion. The key point is that the map $L(f,p)=log(f)+p log(1-f)$ is convex-concave. The way how it is written now could lead to the misunderstanding that it is a convex-concave function of x, which makes no sense of course. As a side remark, there is a typo $L_{x,x}$ should read as $L_{x,x’}$. Even more so, the definition proposed makes only sense for discrete distributions. Otherwise, the odds-ratio should be replaced with the Radon-Nikodym derivative. I also think that absolute continuity of the measures is needed.

	•	I do not understand Example 4. In which sense do you want to approximate the random variable $x$. Almost surely? If so, with respect to which measure? How is $p_{\theta}$ exactly constructed, and what should $f_{\phi}$ be?

	•	In Proposition 1: Do you mean s is drawn from the product measure?

	•	Compactness in Assumption 1 is a strong assumption. You work on a Banach space.

	•	I am missing a proof of Proposition 2 and 3. In my opinion this is not as obvious as you work on a Banach space of functions. The topological details would be quite important to see here. At the current stage it looks to me like a heuristic. If it is a standard result, then at least you should provide references.

	• Explain eq. (13) better. I guess you want to say that your have one data point x and x' is the generator's realization?

	•	Section 3.4: What is $F_{\theta}\times G_{\phi}$? I guess you want to say that you measure distance in the parameter space via the $L^{2}(p)$ distance in the function space. What is $p$? Is it important for the metric? At least you should mention the domain from which mappings $F_{\theta}$ and $G_{\phi}$ are drawn. This is of course important to make these definition meaningful.

	•	Section 6: It would be nice to have some (perhaps speculative) discussion why GDA does perform worse than natural gradient in this example. Has it something to do with the fact that the game of choice has a unique interior equilibrium? Also, what would happen if the equilibrium is at the boundary? In my understanding the dynamics would not be defined on the boundary, so what can be said here?

	•	Proposition A.1: What is f? I guess the potential, but it is not explained. In any case, it is not needed in this Proposition.

	•	Proposition A.2 is problematic: First the partial derivative of \Psi does not exist as written here, unless a proper extension is defined. The way it is currently written, it suggest that it is defined on the probability simplex, which is an affine space for which only directional derivatives can be defined. The notation is also very confusing and it seems that it has not been polished carefully enough.

	•	The $N$ is reserved for the natural numbers, sorry. The current notation is absolutely unacceptable.

	•	Proposition 1 uses notation that has not been defined before. In any case, the result is classical; See e.g. the textbook by Sandholm „Population games and evolutionary dynamics“, a standard reference in game theory. In general, the Lyapunov functions proposed here can also be found in that reference, as well as the invariance functions for matching pennies (and other zero sum games).

	•	Lemma 1 should be $m_{1}$ and $m_{2}$.



**Summary Of The Paper:**

This paper follows a recent and interesting line of research on strategic approaches to GAN training in which the optimization problem of the players is lifted to the space of models employed by the discriminator and generator. The main aim of this literature is to shed light to the question why simple first-order methods perform quite well in this application, although the basic hypothesis (such as monotonicity of the variational inequality) are not very likely to be satisfied. The price to pay via this lifting, due to Gidel, G., Balduzzi, D., Czarnecki, W., Garnelo, M., & Bachrach, Y. (2021, March). A Limited-Capacity Minimax Theorem for Non-Convex Games or: How I Learned to Stop Worrying about Mixed-Nash and Love Neural Nets. In International Conference on Artificial Intelligence and Statistics (pp. 2548-2556). PMLR, is that the saddle-point problem is defined in a Banach space of functions and measures. This paper proposed natural gradient training parameters defining the models employed by the discriminator and generator, respectively.

**Summary Of The Review:**

The ideas driving this paper are interesting, but the mathematical rigor is somewhat questionable. There are several points, partially detailed below, where either the presentation is unclear because of many flaws and typos in the writing, or at least not rigorously established. Overall I appreciated the contribution, but consider it as a borderline.

---

### Official Review · Reviewer_5G2k · 2021-11-02

**Correctness:** 4
**Technical Novelty And Significance:** 2
**Empirical Novelty And Significance:** 3
**Recommendation:** 6
**Confidence:** 3

**Main Review:**

Understanding nonconvex-nonconcave games is pretty important, here the topic is pretty interesting. The paper is basically clearly written, while sometimes hard to follow because of the notations.

My main concerns are as follow:
1. Compare with existing works. A closely related work should be the analysis of GDA in HCC games (Flokas et al. 2021), which authors also compare in experiments. Currently the theorem seems to suggest that natural gradient also attains a Lyapunov function, compared to GDA. It would be better if authors add more discussion on the difference compared to GDA. Currently I found that the Lyapunov function of natural gradient is different from that in (Flokas et al. 2021), and easier, is it the benefit in the analysis?

2. By changing to natural gradient, it should be expected that the convergence will be faster compared to GD, with a cost of inversing matrices. Here authors provide some evidence in experiments. Is there any theoretical evidence to justify the claim that natural gradient is faster than GDA?

Some minor thoughts:
1. I thought the terminology "modified version of Gradient Descent-Ascent" is a little misleading. At first I thought it will be some common variants of GDA, e.g., EG, OGDA... While here it is natural gradient flow, which requires a matrix inversion.

Generally I think the paper is interesting, but it would be better if authors can provide more insights on it. I will appreciate the authors to address my confusions, and definitely reconsider my decision. Thank you very much.

**Summary Of The Paper:**

The paper studied the dynamics of a natural gradient flow in hidden convex-concave problems, and showed that the natural gradient flow attains global convergence to stationary points.

**Summary Of The Review:**

Interesting results while it would be better to add more discussions.

---

### Official Review · Reviewer_Wrtu · 2021-11-02

**Correctness:** 4
**Technical Novelty And Significance:** 3
**Empirical Novelty And Significance:** 2
**Recommendation:** 6
**Confidence:** 4

**Main Review:**

The paper focuses on the particularly interesting class of games known hidden convex-concave games. As the authors point out, this class encompasses many important applications within the setting of min-max optimization, including adversarial example games and GANs. A unifying treatment of such settings is particularly compelling, and a very promising setting. Nonetheless, overall the paper fails to convince about the originality and the significance of the results.

First of all, on of the key points stressed by the authors is that they establish global convergence guarantees. However, it is unclear how this result relates and compares to the existing ones in the context of HCC; e.g. Flokas et al. Solving Min-Max Optimization with Hidden Structure via Gradient Descent/Ascent. The latter paper also presents global convergence results, and a more detailed discussion/comparison with these results is missing. Another caveat is that the analysis they present does not offer a rate of convergence, while as the authors acknowledge the studied dynamics appear to not scale in practice as the involve computing (pseudo)inverses of matrices at every iteration. There have been some techniques to address such issues in the literature (e.g. using sketching), so the authors should definitely consider addressing this intractability issue. A minor unsatisfactory issue is that the analysis is performed in continuous-time and not in discrete-time, but I should note that this is common in this line of work.

With regards to Section 5, the persistence of limit cycles in such settings is a very well-known result. The authors make a further fine-grained characterization of the dynamics, using invariant function analysis, but the importance of such a characterization is not clear; perhaps the authors could explain this point further. It is also perhaps unsatisfactory that the results in Section 5 are only applicable for the specific example of the hidden matching pennies game. Finally, the experimental results are somewhat weak and have only been applied for toy examples. Overall, the practical importance of the results is questionable.

Post Discussion: I thank the authors for the detailed response which addressed some of my concerns/misunderstandings. I would recommend clarifying further the results of Section 5 for the revised version. The importance of these results and a comparison with existing works appears can be further highlighted.

**Summary Of The Paper:**

The authors study a generalized Gradient Descent/Ascent flow in a class of non-convex non-concave games known as hidden convex-concave (HCC) games. Their first contribution is to establish a global convergence guarantee to stationary points. Moreover, for an important instance of the proposed dynamics they characterize the limit cycles using invariant function analysis. Empirical results are also presented to support the theoretical findings.

**Summary Of The Review:**

Overall, the authors study a very interesting emerging setting, covering many important applications. However, the progress appears to be only incremental, and comparisons with existing results appear to be missing.

---

### Official Review · Reviewer_PUzM · 2021-11-02

**Correctness:** 4
**Technical Novelty And Significance:** 2
**Empirical Novelty And Significance:** 2
**Recommendation:** 6
**Confidence:** 4

**Main Review:**

strengths:

The observation that a lot of practical problems including GANs and AEGs are HCC games is interesting and useful, which motivated the proposed NHG method in this work.

The proposed NHG is guaranteed to converge to a Nash equilibrium in the HCC game setting and it can be used to explain cycling behaviour of RD by using a certain metric.

The presentation of results and ideas is clear and easy to understand.

weakness:

I am concerned the proposed NHG method is not scalable and could not be extended to other non-convex settings (like neural networks and GANs as mentioned in the examples).

1. The proposed NHG method is a preconditioning gradient update such that $F$ and $G$ are updated following standard gradient descent ascent update. This would require the preconditioning matrix (e.g., pseudo inverse Fisher) be able to recover from the gradient of $\theta$ the information in the gradient of $F$ and $G$. This is possible if $\theta$ are not low dimensional comparing with $F$ and $G$. However, suppose we use some compact feature and low dimensional $\theta$, then could the author comment how this is achievable (preconditioning low dimensional gradient of $\theta$ can behave like doing gradient update on $F$ and $G$)?

2. As also noted by the authors, the preconditioning requires expensive computational expenses, which seems to be a problem if we have neural networks weights as $\theta$.




**Summary Of The Paper:**

This paper studied a non-convex non-concave Zero-Sum game setting called Hidden Convex-Concave game (HCC), as shown in Def. 1, which includes several examples, such as Hidden Matching Pennies games, Generative Adversarial Networks (GANs) / WGANs, and Adversarial Example Game (AEG). The observation is that although the min-max objective is non-convex non-concave w.r.t. the parameters (e.g., weights of neural networks), it could be convex concave of the outputs/strategies of the players $F$ and $G$.

Under Assumption 1 of there exists a Nash equilibrium $(F^*, G^*)$, the authors studies/proposed an algorithm called Natural Hidden Gradient flow (NHG), which is a preconditioned gradient flow update, such that $F$ and $G$ are doing normal Euclidean gradient descent because of the HCC game is convex concave w.r.t. $F$ and $G$.

The authors first showed that for a one-dimensional and finite-sum of convex-concave objectives, the distance to optimum is a Lyapunov function of NHG, and thus NHG converges to the Nash equilibrium.

The authors then connected NHG to replicator dynamics (RD) in literature, under a certain metric of geometry. They showed that in a 2-Team Hidden Matching Pennies game, there exist several invariant functions, which explains why RD manifest cycling orbits.

Finally, the authors conducted simulations to verify the proposed NHG method, showing that it is more stable than Gradient Descent-Ascent (GDA).

**Summary Of The Review:**

Overall, this paper makes an interesting observation about the HCC game setting and proposes a reasonable NHG method, reducing the non-convex non-concave problem to a convex-concave problem. However, I am concerned with the preconditioning in the NHG method, making it not scalable and potentially not possible to be used in other non-convex settings. Therefore, I make a borderline decision for the current moment.

---

### Decision · Program_Chairs · 2022-01-20

**Decision:**

Accept (Poster)

**Comment:**

The paper studies the convergence of a generalized gradient descent ascent (GDA) flow in certain classes of nonconvex-nonconcave min-max setups. While the nonconvex-nonconcave setups are computationally intractable and GDA is known to converge even in some convex-concave setups, the paper argues that (generalized) GDA can converge on what is dubbed "Hidden Convex-Concave Games" in the paper and argued that it contains GANs as a special case.

The reviewers all found the paper interesting and a worthy contribution to the literature on nonconvex-nonconcave zero-sum games. Main concerns expressed by the reviewers were w.r.t. the lack of convergence rate established for the considered dynamics, novelty compared to existing work, and practical usefulness of the considered scheme, as it involves preconditioning/matrix inversion. The authors made an effort to address all the concerns, to the extent possible.

Given the complexity of nonconvex-nonconcave min-max setups, their importance in GAN training, and the insightful perspective on hidden convexity/concavity in typical problem instances, I would like to see this paper published at ICLR. I would, however, strongly advise the authors to take all of the reviewers' comments into account when preparing a revision.